# Arti-PG: A Procedural Toolbox to Synthesize Large-Scale and Diverse Articulated Objects with Rich Annotations

## Abstract

The acquisition of substantial volumes of 3D articulated object data is expensive and time-consuming, and consequently the scarcity of 3D articulated object data becomes an obstacle for deep learning methods to achieve remarkable performance in various articulated object understanding tasks. Meanwhile, pairing these object data with detailed annotations to enable training for various tasks is also difficult and labor-intensive to achieve. In order to expeditiously gather a significant number of 3D articulated objects with comprehensive and detailed annotations for training, we propose **Arti**culated Object **P**rocedural **G**eneration toolbox, *a.k.a.* **Arti-PG** toolbox. Arti-PG toolbox consists of i) descriptions of articulated objects by means of a generalized structure program along with their analytic correspondence to the objects' point cloud, ii) procedural rules about manipulations on the structure program to synthesize large-scale and diverse new articulated objects, and iii) mathematical descriptions of knowledge (*e.g.* affordance, semantics, *etc.*) to provide annotations to the synthesized object. Arti-PG has two appealing properties for providing training data for articulated object understanding tasks: i) objects are created with unlimited variations in shape through program-oriented structure manipulation, ii) Arti-PG is widely applicable to diverse tasks by easily providing comprehensive and detailed annotations. Arti-PG now supports the procedural generation of 26 categories of articulate objects and provides annotations across a wide range of both vision and manipulation tasks, and we provide exhaustive experiments which fully demonstrate its advantages. We will make Arti-PG toolbox publicly available for the community to use. More details, analysis and discussions are provided in technical appendices.

## 1 Introduction

Articulated objects, comprised of rigid segments interconnected by joints that enable translation and rotation movements, play an important role in daily life. Learning to understand articulated objects is an essential topic in a wide range of research areas, including computer vision, robotics and embodied AI. In the current data-driven era, the availability of a large amount of training data has become indispensable for the successful implementation of deep neural networks to understand articulated objects.

Common 3D articulated object data acquisition methods are either designing 3D CAD models by artists (Chang et al., 2015; Xiang et al., 2020) or scanning real-world objects using scanners (Liu et al., 2022)[1], both of which have huge demands on time and money. Furthermore, comprehensive and detailed annotations are required for these object data to support training in various articulated object understanding tasks, which are also challenging to obtain. As a result, the issue of data scarcity is observed across different tasks supported by existing datasets (Mo et al., 2019; Liu et al., 2022), limiting the power of deep neural networks to comprehensively analyze and model articulated objects. Given that prior research has examined little on how to mitigate this issue, it remains a pressing problem that requires attention.

---

[1]Here, we discuss about how the data are created from scratch, since it is usually unavailable to collect data from the Internet for novel categories in real-world applications.

In this paper, we propose **Arti**culated Object **P**rocedural **G**eneration toolbox (**Arti-PG** toolbox) as a solution to this issue, which aids in expeditiously gathering a significant number of 3D articulated objects with rich annotations. Arti-PG is developed based on the idea of procedural generation (Togelius et al., 2014), referring to synthesizing data with generalized procedural rules.

Inspired by research in visual cognition and brain science (Habel & Eschenbach, 2006; Ullman, 2000; Palmeri & Gauthier, 2004; Biederman, 1987), we assume that a 3D object can be properly described as the combination of a macro spatial structure and micro geometric details. By first describing an articulated object's spatial structure as generalized programs and geometric details as point-wise correspondence between the object's point cloud and structure, novel 3D articulated objects can be synthesized in two steps: i) create a variation of the structure via the application of randomized mathematical rules to the programs, and ii) recover the geometric details according to the point-wise correspondence. Subsequently, we are able to automatically assign annotations to the synthesized objects using mathematical descriptions defined upon the structure programs. Such annotated synthesized objects can then be used to enrich the training set for various tasks, facilitating network training.

Therefore, we construct the Arti-PG toolbox with three components: i) structure programs of articulated objects along with their correspondence to the objects' point cloud, ii) procedural rules for structure program manipulation, and iii) mathematical descriptions of knowledge (*e.g.* affordance, semantics, *etc.*) for annotations. Arti-PG now supports 26 categories of articulate objects that are most commonly seen and provides different kinds of knowledge for a wide range of tasks. Users can easily use the codes in the toolbox to synthesize large-scale and diverse articulated objects with rich annotations to train their models.

Our procedural approach has the following appealing properties. 1) **Program-oriented Structure Manipulation**: Training set can be significantly enriched by synthesizing objects with unlimited variations in shape through alterations of the structure program. Such alterations can be automatically generated via randomized mathematical rules. 2) **Analytic Label Alignment**: Comprehensive and detailed annotations of various types can be mathematically defined in the structure program, after which they can be analytically aligned with the synthesized object.

Benefiting from these properties, Arti-PG holds advantages in terms of the diversity of generated objects, applicability to a wide range of tasks and effectiveness in solving data scarcity. Compared to data augmentation methods which also increase the diversity of training data but cannot freely assign labels to them and hence are limited to specific tasks, Arti-PG is applicable in different tasks and therefore distinguishes itself from conventional data augmentation methods.

We have collected a total number of 3096 3D articulated objects across 26 categories with complex shapes from influential and open-source datasets (Yi et al., 2016; Mo et al., 2019; Xiang et al., 2020) to evaluate our approach. In the following sections, we will fully demonstrate the mechanism of our approach and further showcase the superiority of Arti-PG through evaluations from both vision and robotic aspects: part segmentation, part pose estimation, point cloud completion and object manipulation.

## 2 BACKGROUND AND MOTIVATION

### 2.1 ARTICULATED OBJECT DATASETS

The enormous advancement of machine learning is accompanied by the vigorous development of large-scale datasets across various modalities. Although large datasets (Chang et al., 2015; Deitke et al., 2023; Lin et al., 2015) have appeared in research areas such as images and rigid shapes, it is much more costly and laborious to acquire articulated object data as well as annotations for various articulated object understanding tasks (Liu et al., 2022; Xiang et al., 2020; Wang et al., 2019). Therefore, there are not many large-scale articulated object datasets that have been proposed (Jiang et al., 2022; Mao et al., 2022; Wang et al., 2019; Liu et al., 2022; Xiang et al., 2020). One of the most commonly used dataset, PartNet-Mobility Xiang et al. (2020), offers 2,346 object models from 46 common indoor object categories, about only 50 objects per category on average. All the object models are collected from 3D Warehouse, a 3D model library containing CAD models of real world brands promoting products designed by experts.

## 2.2 ARTICULATED OBJECT UNDERSTANDING TASKS

Articulated objects play an important role in human daily life and understanding these objects is crucial for machine intelligence to perceive and interact with them. To fully understand articulated objects, a series of vision and manipulation tasks have been studied.

**Vision Tasks.** Part segmentation, part pose estimation and point cloud completion are three important vision tasks for articulated object understanding. Part segmentation (Qi et al., 2017a;b; Guo et al., 2021; Zhao et al., 2021), which is one of the most fundamental tasks, assigns a semantic label to each point of the object. Part pose estimation (Geng et al., 2023; Liu et al., 2023) involves querying the 7-dimensional transformation of detected parts on the object, including the scale, rotation and location of the parts. In these tasks, it is critical to have a good understanding of the spatial structure of an object. On the other hand, point cloud completion aims to estimate the complete shape of objects from partial observations (Yuan et al., 2018; Tchapmi et al., 2019; Wen et al., 2020; Xiang et al., 2022), which pays more attention on the geometric details.

**Manipulation Tasks.** Articulated object manipulation is a set of various tasks focusing on how an embodied agent properly interacts with articulated objects (Geng et al., 2023; Mo et al., 2021; Wang et al., 2022; Ning et al., 2024). For example, Where2Act (Mo et al., 2021) proposed to predict per-pixel action likelihoods and proposals for manipulation. Where2Explore (Ning et al., 2024) proposed a few-shot learning framework for articulated object manipulation that measures affordance similarity across categories to migrate affordance knowledge to novel objects. GAPartNet (Geng et al., 2023) released a dataset with semantic and affordance labels and proposed a manipulation pipeline by leveraging the concept of actionable parts. The success rate of manipulation using these proposals largely depends on the understanding of affordances on articulated objects.

In this paper, we will conduct exhaustive experiments on the four listed tasks to comprehensively evaluate the quality of our synthetic training data in terms of spatial structure, geometric details and annotations, and also demonstrate the wide applicability of our approach.

## 2.3 SCARCITY OF TRAINING DATA IN ARTICULATED OBJECT RESEARCH

In the era of deep learning, a sufficient amount of training data is crucial for neural networks to achieve remarkable performance. However, in the field of articulated object research, the scarcity of training data remains a major obstacle for various articulated object understanding tasks. The challenge in object acquisition is one of the major reasons for data scarcity. When collecting 3D articulated object data of novel categories, common practices would be to design CAD models or scan real-world objects, both of which can be costly and time-consuming. Specifically, designing one CAD model from scratch would generally require a specialized artist to spend more than 2 hours while the corresponding fees can exceed $100 (Liu et al., 2022). On the other hand, for scanning objects, the high expenses associated with acquiring the scanner and numerous real-world objects, including high-value items like washing machines, also cannot be neglected. Meanwhile, the difficulties in data annotation further restrict the applicability of existing object data. Generally, manually annotating a 3D shape involves viewing it on a 2D screen, which would require the annotator to constantly change viewing angles to complete the annotation. Furthermore, some types of annotations such as affordances for manipulation are extremely complicated to manually annotate (Mo et al., 2021), resulting in few existing datasets available for affordance labels. Apart from the above points, it is also challenging to comprehensively label an articulated object to support a wide range of tasks, such as semantics, 6-dof pose, grasp pose, *etc*.

Unfortunately, few researchers have focused their attention on directly addressing the data scarcity problem. Yet some previous studies on data augmentation (Chen et al., 2020; Li et al., 2020; Kim et al., 2021; Lee et al., 2021) can be applied in this context to alleviate the impact of data scarcity, leveraging their power to enhance the diversity of training data and prevent models from overfitting. For example, PointMixup (Chen et al., 2020) proposed a technique of interpolation between existing point clouds. PointWOLF (Kim et al., 2021) applied smoothly varying non-rigid deformations to the point clouds for diverse and realistic augmentations. However, this line of works cannot provide additional annotations for the augmented data unless they already exist in the original data, which restricts the augmented data to specific object modeling tasks.

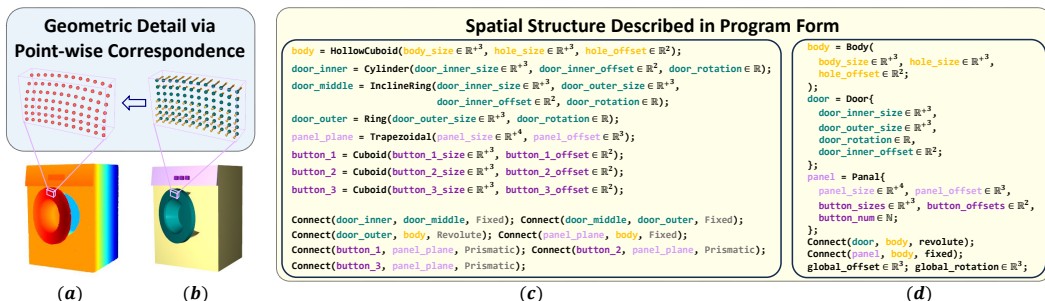

Figure 1: **a.** The point cloud of a washing machine. A small area of its door surface is zoomed in for a clear view of geometric details. **b.** Describing the object with spatial structure (bottom) and geometric details (top). The brown arrows concretely represent point-wise correspondence between points of the structure and the real point clouds. **c.** Naive program description of the structure in (b). The correspondence between the program and structure is indicated by the same color. Elementary primitive templates are in black font (*e.g. Cylinder*) and instances of elementary primivies are in colored font (*e.g. door_inner*). **d.** Program description of the structure in (b) via advanced primitive template. Advanced primitive templates are in black font (*e.g. Body*) and instances of advanced primitives are in colored font (*e.g. body*).

## 3 ARTI-PG: METHODOLOGY

### 3.1 OVERVIEW

The research in visual cognition and brain science (Humphreys et al., 1999; Habel & Eschenbach, 2006; Ullman, 2000; Palmeri & Gauthier, 2004; Biederman, 1987; Hummel & Biederman, 1992) shows that the perceptual recognition of objects by human is conceptualized to be a process in which the spatial properties of the object are segmented into an arrangement of simple geometric primitives such as cuboids and spheres. Inspired by this point of view, we assume that an object in 3D space can be properly represented with a macro spatial structure and its micro geometric details. Fig. 1 gives a brief illustration.

The macro spatial structure of an object includes aspects of the geometric primitives and the connectivity relationships among them. By describing the primitives as i) specific shapes along with corresponding geometric parameters and ii) their connectivity relationships as relative constraints in DoF (degree of freedom), the structure of an object can be represented quantitatively. Then we can further consider the micro geometric details as shape deformation on the geometric primitives within the macro structure.

Intuitively, each primitive can be perceived as a class template which creates shape instances with specific parameters, and the connectivity relationships can be defined as binary descriptors given two shape instances. Based on this observation, we formulate the structure of an object as a program-like representation in our implementation, where generalized geometric primitives and common connectivity relationships are mathematically defined. To formulate the deformation for the geometric details, we find the point-wise correspondence between the object's point cloud and the points on each primitive's surface and describe the deformation as the transformation of each pair of points, drawing inspiration from the idea in BPS (Prokudin et al., 2019).

After representing an object with its structure program and geometric details as aforementioned, infinite new objects with unlimited variations in shape can be synthesized through i) alterations of the program via generalized procedural rules and ii) recovering the geometric details according to the point-wise correspondence. Given that the entire program is mathematically defined, we can easily describe different types of annotations on the program using mathematical descriptions and analytically align them to the synthesized objects. In this manner, numerous new objects with rich annotations can be effortlessly obtained.

In the following sections, we first introduce how to represent an object asset with a structure program and geometric details in Sec. 3.2 and Sec. 3.3, and then demonstrate the procedural generation rules

in Sec. 3.4 and Sec. 3.5. Finally, Sec. 3.6 shows the process of label alignment. Please refer to Appendix A-E and H for comprehensive implementations and discussions of technical details.

## 3.2 Program Description of Spatial Structure

In our approach, the spatial structure of an object, including parameterized geometric primitives and connectivity relationships, is described in program form. Considering that each type of geometric primitive represents a group of shapes that share the same properties, we design each geometric primitive as a single class template, whose constructor depicts its general geometric properties. By assigning corresponding parameters, the constructor will instantiate a specific shape of this primitive. The parameters include intrinsic ones describing the geometric attributes like *height and radius of a cylinder*, and extrinsic ones like *positions and orientations of the whole shape*. The connectivity relationship, as the other component in the structure program, is designed as a binary descriptor. It describes how two shape instances are physically connected, by imposing mathematical constraints between them which reduce the total DoF. Fig. 1-c provides an example of a program description for the spatial structure in Fig. 1-b.

Class templates of elementary primitives, like *cuboid* and *cylinder*, are initially designed from scratch. Observing that common real-world objects within a category often exhibit a consistent hierarchy in structure (Ullman, 2000; Mo et al., 2019; Wang et al., 2011), we further introduce advanced primitive templates to capture the structural regularities of components in a high-level hierarchy of an object category.

An advanced primitive template is constructed based on a set of elementary primitives with specific spatial layouts and their connectivity relationships. We additionally introduce discrete intrinsic parameters in an advanced template to describe regular repetitions of certain elementary primitives. Given that there are naturally different types of structural regularities for high-level hierarchical components, we present multiple advanced primitive templates with various designs to cover the diversity. After introducing advanced primitives in the structure program, the program can better reflect the arrangement and relations between shape parts and be more concise, see Fig. 1-d.

To efficiently and effectively obtain the structure program of a real object, we have elaborately designed a user-friendly structure program annotation system for guidance. Due to space limitations, we introduce the structure program annotation system in Appendix E and provide a video demonstration in the supplementary material.

## 3.3 Geometric Detail via Point-wise Correspondence

After the macro spatial structure of the object is properly represented, we discuss how to formulate the micro geometric details in this section. We describe the geometric details with a set of point-wise correspondences between the structure and the object which depict a 3D deformation on point clouds. By applying the deformation to the point cloud of the structure, we will get a new point cloud that fully represents the object.

Specifically, let $X = \{\mathbf{x}_i \in \mathbb{R}^3 | i \in [1, n]\}$ be the point cloud uniformly sampled from the visible surface of the shape described by the structure program[2], $Y = \{\mathbf{y}_i \in \mathbb{R}^3 | i \in [1, m]\}$ be the point cloud of the object itself. Our goal is to find a deformation $\Delta X = \{\Delta \mathbf{x}_i \in \mathbb{R}^3 | i \in [1, n]\}$ from $X$ to $Y$ with minimum cost, written by

$$\min_{\Delta X} \quad \frac{1}{n} \sum_{i=1}^{n} ||\Delta \mathbf{x}_i||_2 \tag{1}$$
$$s.t. \quad \forall i \in [1, n], \ \mathbf{x}_i + \Delta \mathbf{x}_i \in Y$$

where $\Delta \mathbf{x}_i$ is the correspondence vector for point $\mathbf{x}_i$, and $\mathbf{x}_i + \Delta \mathbf{x}_i$ indicates which point in Y corresponds to $\mathbf{x}_i$. Inspired by BPS (Prokudin et al., 2019), Eq. 1 can be solved as

$$\Delta X = \{\Delta \mathbf{x}_i = \arg\min_{\mathbf{y}_j \in Y} ||\mathbf{x}_i - \mathbf{y}_j||_2 - \mathbf{x}_i \mid i \in [1, n]\} \tag{2}$$

---

[2]Here the points are analytically bounded to the geometric primitives, that is, the positions of the points are all analytic functions of the structure's parameters. For example, the position of a point on a sphere in its local coordinate system can be calculated as $(r \sin \theta \cos \phi, r \sin \theta \sin \phi, r \cos \theta)$, where $r$ is radius and $\theta, \phi$ are the polar and azimuthal angles respectively.

Then we can use $X' = \{\mathbf{x}'_1..., \mathbf{x}'_n\}$ to denote the geometric details on the structure representation where $\mathbf{x}'_i = \mathbf{x}_i + \Delta\mathbf{x}_i{}^3$.

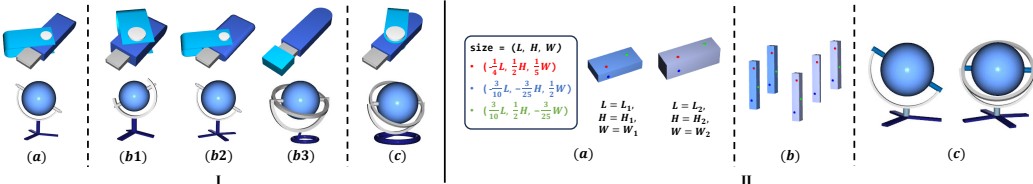

Figure 2: Fig. I illustrates examples of structure manipulation. I-(a): The original structure. I-(b1-b3): Structures after being manipulated by CPA, DPA, APA respectively. I-(c): Structure after being manipulated by the combination of three alterations. Fig. II shows examples of mapping between points in CPA (a), DPA (b) and correspondence between elementary primitives in APA (c). In II-(a) and II-(b), points are analytically bounded to the primitive with parameterized coordinate representation. II-(c) depicts correspondence between elementary primitives by the same colors, such as silver bracket in both globes.

## 3.4 PROGRAM-ORIENTED STRUCTURE MANIPULATION

So far, we have discussed how to represent a given object with our structure program and geometric details. In this section, we delve into the process of manipulating the original structure of a given asset to create diverse new structures. We design generalized procedural rules which encompass different perspectives of the structure program's alterations, including continuous parameters, discrete parameters and advanced primitives. Fig. 2 illustrates examples of new structures after manipulation.

**Continuous Parameter Alteration (CPA).** Apply random perturbations to the continuous parameters of primitives in the structure program. Some of the continuous parameters are automatically adapted rather than being perturbed due to constraints imposed by connectivity relationships. Such constraints ensure the generated structure to be stable and valid, meaning that there are no primitive collisions or floating elements. As shown in Fig. 2-I-(b1), the sizes of primitives and the angles between them are perturbed in this process.

**Discrete Parameter Alteration (DPA).** Apply random changes to the discrete parameters of advanced primitives within a reasonable range. This will vary the total amount of elementary geometric primitives in the structure program and thereby change the complexity of the whole structure. As shown in Fig. 2-I-b2, the number of arc sides on the USB body and legs of the globe base are increased through DPA.

**Advanced Primitive Alteration (APA).** Randomly replace an advanced primitive with another that represents the same hierarchical component. This will significantly diversify the structure of synthesized objects. We let the replacement primitive inherit the overall dimensions of the replaced one so that it stays in proportion to other primitives in the structure. Additionally, APA will also make random alterations on the existence of non-essential high-level hierarchical components. As shown in the example of Fig. 2-I-b3, the rotated cap and the rounded rectangle body in the original USB are manipulated into a detached cap and a round tailed body. The bracket of the globe becomes more complex and the legged base is altered to a ring base.

We adopt the procedural rules in the order of APA, DPA, CPA with the aim of creating a wide variety of new structures. Considering that the randomness introduced in these procedural rules may lead to the occurrence of extreme parameters, the shape described by the structure program with such extreme parameters will occasionally deviate from physical laws to some extent, *e.g.* collision between two primitives. To this end, we design an exception handling module to verify the validity of the structure program. This module will monitor the alternation process and automatically locate and adjust the erroneous parameters. In Appendix H, we provide detailed examples of 'globe base' to better demonstrate structure manipulation with more details.

---

[3]Note that the points in $X'$ is one-to-one correspondent to the points in $X$, hence they are also analytically bounded to the geometric primitives.

## 3.5 RECOVERY OF GEOMETRIC DETAILS

Now we discuss how to recover the geometric details for a new structure by migrating the geometric details from the original object. Intuitively, given that the geometric details are analytically bounded to the geometric primitives in a structure as discussed in Sec. 3.3, the migration can be carried out by finding the mapping between points from surfaces of the original and the new structures, *i.e.* before and after the three kinds of alterations. **1) CPA:** Since the surface points are analytically bounded to the primitives, the mapping is automatically built according to the primitives' parameters. **2) DPA:** As the value of discrete parameter reduces, primitives are removed and the mapping can be ignored. Oppositely, primitives are added via replication and the mapping is automatically built among the repeated primitives. **3) APA:** We assign correspondence between the elementary primitives in the original and altered advanced primitives based on their hierarchical consistency, to simplify the mapping from the advanced primitive level to the elementary primitive level. If two corresponding elementary primitives belong to the same template, their mapping is built as discussed in CPA. Otherwise, their mapping is built by map projection techniques (Snyder, 1987), Examples are provided in Fig. 2-II.

After finding the mapping, there are two issues that should be further dealt with. i) Only the points on visible surfaces are covered by geometric details in the original object. Noticing that some points on the invisible surfaces of the original structure may become visible after structure modification, these invisible points should also be covered by geometric details. Therefore, we complete the geometric details separately for each elementary primitive, by duplicating the visible points to invisible areas based on the properties of the primitive's local geometric patterns such as translational and rotational symmetry. ii) The geometric details in Eq. 2 are in the world coordinate system, which implies that they cannot be directly used for migration as the normal direction of mapped points may be changed. To this end, we transform each $\Delta \mathbf{x}_i$ to a new vector $\Delta \hat{\mathbf{x}}_i$ relative to the point normal at $\mathbf{x}_i$.

Finally, we recover the geometric details for the new structure by i) assigning relative geometric details (*i.e.* $\{\Delta \hat{\mathbf{x}}_i\}$) to the points on the visible surface of the new structure according to the mapping, and ii) transforming the relative geometric details back to the world coordinate system according to the point normal.

## 3.6 ANALYTIC LABEL ALIGNMENT

As described in previous texts, we are able to synthesize a new object according to the altered structure program and geometric details, and each point of the new object is analytically bounded to the geometric primitives in the structure program. Taking advantage of this property, we can analytically align knowledge labels to the object's point cloud.

Specifically, we assign the labels onto the geometric primitives using functions defined on parameters of the primitives. This allows for the automatic labeling of spatial structures when they change with the variation of parameters. Fig. 3 shows examples of labeling on structures, including the center of ring handles, the outer edge of doors and the rim on knobs, these labels provide affordances for interaction. Then, through the point-wise correspondence of geometric details, the labels on the structures can be further automatically propagated to the point clouds of generated objects. Following such approach, we are able to synthesize a wide array of labeled objects without additional human effort.

## 4 ARTI-PG: TOOLBOX

Following our Arti-PG methodology, we construct the Arti-PG toolbox to facilitate the community easily and expeditiously synthesizing large-scale articulated object data for training using our approach. The toolbox consists of three important components: i) Off-the-shelf primitive templates for each object category, and also abundant structure program descriptions and point-wise correspondences for different articulated objects; ii) Procedural programs for structure manipulation, as well as codes for geometric detail recovery; iii) Programs of different kinds of knowledge definition along with the codes for analytic label alignment on procedurally generated objects.

Particularly, our toolbox now covers 26 categories of articulated objects which are widely used in vision and manipulation tasks (Xiang et al., 2020; Mo et al., 2021; Zhao et al., 2021), along with

Figure 3: Illustrations of analytically assigning labels on spatial structures of various categories with functions (described in mathematical formulas, the coordinate center is indicated by the arrow, zoom in for a clear view). We take affordable areas that are reasonable to interact with the object as examples of labels. **a.** edge of microwave door. **b.** lower half of handle (we can still represent such area with same parameters and functions even if the handle is rotated). **c.** area between supporting parts on the handle and the top rim of cap knob. **d.** the top rim of cap knob and the center of kettle ring handle.

structure program descriptions of 2133 objects from Mo et al. (2019); Xiang et al. (2020) which contain complex spatial structures, available for diverse procedural generation results.

With the codes and data in the toolbox, it is very easy for users to synthesize new articulated objects, by i) applying the codes for structure program manipulations to structure descriptions of certain objects, ii) performing the codes for geometric detail recovery according to the point-wise correspondence of the objects, and iii) conducting analytic label alignment with programs of different kinds of knowledge definition. The purpose of us proposing Arti-PG toolbox is to help researchers effortlessly acquire a large amount of well-annotated data to meet their research needs in specific applications about articulated objects.

## 5 EXPERIMENTS

We thoroughly evaluate the effectiveness of our approach in synthesizing high-quality and richly-annotated articulated objects for training deep neural networks in both vision and manipulation tasks. The vision tasks include part segmentation, part pose estimation and point cloud completion. The manipulation tasks focus on guiding an embodied agent to properly interact with articulated objects.

From widely-used datasets (Yi et al., 2016; Mo et al., 2019; Xiang et al., 2020), we gather 3096 articulated objects spanning over 26 categories with varying structures to support the evaluation across the aforementioned tasks. We only use the objects in Arti-PG toolbox for procedural generation that are in the training set for all these tasks.

Representative approaches for each task (Zhao et al., 2021; Xiang et al., 2022; Mo et al., 2021; Ning et al., 2024; Geng et al., 2023) including state-of-the-art are adopted as baselines to evaluate the improvement achieved after being assisted by our synthesized data and annotations. The training is conducted on randomly synthesized new objects and stops when the training loss converges. In the following sections, we present the main results and analysis for each task. Please refer to Appendix F for more details, results, comparisons and discussions of our experiments, and Appendix G for visualizations of our synthesized objects.

### 5.1 VISION TASKS

In this part, we first introduce details about the experiments on three important vision tasks, part segmentation, part pose estimation and point cloud completion, and then discuss about the results of these experiments together.

**Part Segmentation.** We follow the part definition proposed by Mo et al. (2019); Xiang et al. (2020) as the ground truth labels for part segmentation, and obtain the part labels for our synthesized training objects by first assigning each primitive in the structure program its part label, and then propagating such labels to the objects' point cloud. We uniformly sample 2048 points as input. We take the classical and widely-used PointTransformer (Zhao et al., 2021) as baseline network, and compare our approach with PointWOLF (Kim et al., 2021), a point cloud augmentation technique

developed for the task. Mean accuracy (mAcc) and mean IoU (mIoU) are adopted as evaluation metrics following the baseline.

**Part Pose Estimation.** For this task, we refer to NPCS from GAPartNet (Geng et al., 2023) as the baseline, and report metrics including rotation error ($R_e$), translation error ($T_e$), scale error ($S_e$), 3D mIoU, ($5°$, 5cm) accuracy ($A_5$) and ($10°$, 10cm) accuracy ($A_{10}$) following the baseline. The ground truth part pose for our synthesized training objects is obtained by calculating the transformation from the reference coordinate system to the part's coordinate system.

**Point Cloud Completion.** Following Yuan et al. (2018); Xiang et al. (2022), we uniformly sample 16384 points from each object in both training and test sets as the complete point clouds and then acquire partial point clouds by back projecting the complete shapes into 8 different partial views. 2048 points are sampled from each partial point cloud as input. We use SnowFlakeNet (Xiang et al., 2022) as a strong baseline network for evaluation and adopt the Chamfer Distance (CD) between the completed point cloud and the ground truth as metric.

**Main Results.** The main results of the three vision tasks are reported in Tab. 1. Remarkable performance improvements over the baselines are achieved for all tasks under all metrics, with notable improvements of approximately $10\%$ in metrics such as CD, $T_e$, and $S_e$. As these metrics together reflect the understanding of articulated objects in terms of both spatial structure and geometric details, prominent performance on all these metrics indicates that the objects synthesized by our approach possess high quality in both aspects. The comparison with data augmentation technique PointWOLF is also shown in Tab. 1, which demonstrates two benefits of our approach: i) synthesized objects are more effective to improve a model's performance, and ii) our approach is widely applicable to various tasks.

Table 1: Experimental results of part segmentation, part pose estimation and point cloud completion. *Impr.* denotes the improvement of Arti-PG over the baseline in absolute value.

| Tasks | Segmentation | | Part Pose Estimation | | | | | | Completion |
|---|---|---|---|---|---|---|---|---|---|
| Methods | mAcc(%) ↑ | mIoU(%) ↑ | $R_e$(°) ↓ | $T_e$(cm) ↓ | $S_e$(cm) ↓ | mIoU(%) ↑ | $A_5$(%) ↑ | $A_{10}$(%) ↑ | CD($\times 10^{-4}$cm) ↓ |
| × | 89.5 | 74.5 | 11.0 | 0.043 | 0.025 | 44.1 | 24.8 | 51.9 | 11.3 |
| Arti-PG | **91.3** | **79.4** | **10.5** | **0.039** | **0.022** | **48.3** | **25.9** | **53.0** | **10.4** |
| *Impr.* | 1.8 | 4.9 | 0.5 | 0.004 | 0.003 | 4.2 | 1.1 | 1.1 | 0.9 |
| PointWOLF | 89.7 | 75.8 | - | - | - | - | - | - | - |

## 5.2 MANIPULATION TASKS

We now report the performance of manipulation baselines, namely Where2Act (Mo et al., 2021), GAPartNet (Geng et al., 2023), and state-of-the-art Where2Explore (Ning et al., 2024), after using our synthesized data for training. Particularly, the training of Where2Act and Where2Explore rely on affordance labels which are not provided in an articulated object dataset. As a compromise, they explore the affordance labels of an object according to the outcome of simulated interactions, which may result in inaccurate and noisy labels due to imperfections of the simulator. In comparison, when training these frameworks on our synthesized data, we use the high-quality and well-defined affordance labels obtained according to Sec. 3.6, instead of estimating affordances with simulation. As the success of manipulation largely depends on how well a model understands the affordances of the target articulated object, these experiments will substantially prove the quality of the annotations provided by our approach.

**Experiment Settings.** A total of 15 representative categories of objects among PartNet-Mobility (Xiang et al., 2020) are used in experiments. Following Mo et al. (2021), we have removed those that are too small or do not make sense for single-gripper manipulation. A full list of the specific tasks on these objects is provided in Appendix F Tab. 8, which can be categorized into two general action types: pushing and pulling. We follow the baselines for the environment settings and action settings, see Appendix F. Success rate is used as the evaluation metric.

**Main Results.** Tab. 2 highlights great improvements after incorporating our synthesized data for training these baselines, especially for Pull-Where2Explore whose improvement reaches $28\%$. As Where2Act (Mo et al., 2021), Where2Explore (Ning et al., 2024) and GAPartNet (Geng et al., 2023) respectively rely on affordance and part pose labels for training, these results demonstrate the

remarkable capability of our approach to provide high-quality annotations of various types including different kinds of affordable areas and part poses.

Table 2: Experimental results of manipulation tasks. *Impr.* denotes the improvement of Arti-PG over the baseline in absolute value.

| Action Type | Methods | Where2Act | Where2Explore | GAPartNet |
|---|---|---|---|---|
| | × | 21.4 / 7.6 | 25.9 / 9.3 | 26.6 / 12.9 |
| Push / Pull | Arti-PG | **26.4 / 9.2** | **32.8 / 11.9** | **33.5 / 16.5** |
| | *Impr.* | 5.0 / 1.6 | 6.9 / 2.6 | 6.9 / 3.6 |

## 5.3 ABLATION STUDY

**Contribution Analysis.** Arti-PG consists of procedural rules in two aspects, structure manipulation and geometric detail recovery. Tab. 3 provides ablative results about the contribution of these two aspects in the aforementioned tasks. Generally, both aspects contribute to the improvement of all the tasks. In specific, the impact of structure manipulation is more pronounced in part segmentation and part pose estimation while the influence of geometric detail recovery is more prominent to point cloud completion, and their contributions are balanced in more comprehensive tasks, namely manipulation. This finding is consistent with the structure and geometric details biases in these tasks.

**Structure Manipulation Rules.** We further investigate the contribution of the three kinds of structure manipulation rules in Tab. 4. As stronger manipulation rules are introduced progressively, the performance of the networks gradually improves, indicating that these rules can effectively increase the diversity of the synthesized object structures and thus bring better coverage of samples in the test set.

Table 3: Contribution analysis of structure manipulation (M) and geometric details recovery (R).

| Tasks | Segmentation | | Part Pose Estimation | | Completion | Manipulation | |
|---|---|---|---|---|---|---|---|
| Methods | **mAcc**(%) ↑ | **mIoU**(%) ↑ | **mIoU**(%) ↑ | $A_5$(%) ↑ | CD($\times10^{-4}$cm)↓ | push ssr(%) ↑ | pull ssr(%) ↑ |
| × | 89.5 | 74.5 | 44.1 | 24.8 | 11.328 | 21.4 | 7.6 |
| M | 90.6 | 76.7 | 47.0 | 25.3 | 11.105 | 25.6 | 8.7 |
| M + R | **91.3** | **79.4** | **48.3** | **25.9** | **10.408** | **26.4** | **9.2** |

Table 4: Ablation study on three kinds of structure manipulation rules.

| Tasks | Segmentation | | Part Pose Estimation | | Completion | Manipulation | |
|---|---|---|---|---|---|---|---|
| Methods | **mAcc**(%) ↑ | **mIoU**(%) ↑ | **mIoU**(%) ↑ | $A_5$(%) ↑ | CD($\times10^{-4}$cm)↓ | push ssr(%) ↑ | pull ssr(%) ↑ |
| × | 89.5 | 74.5 | 44.1 | 24.8 | 11.328 | 21.4 | 7.6 |
| CPA | 90.2 | 76.5 | 47.5 | 25.5 | 10.961 | 21.8 | 7.9 |
| DPA + CPA | 90.8 | 79.0 | 47.7 | 25.5 | 10.510 | 22.5 | 8.4 |
| All | **91.3** | **79.4** | **48.3** | **25.9** | **10.408** | **26.4** | **9.2** |

## 6 CONCLUSION

In this paper, we introduce Arti-PG toolbox, a procedural generation toolbox aids in synthesizing numerous and diverse 3D articulated objects associated with rich annotations, in order to deal with the data scarcity issue in various articulated object understanding tasks. The novelties of Arti-PG are threefold. First, we propose a program description for macro spatial structure and a point-wise correspondence representation for micro geometric details to mathematically represent the object asset. Second, we design generalized procedural rules to synthesize new objects by first creating a variation of the structure via manipulating the structure program, and then recovering the geometric details according to the point-wise correspondence. Third, we demonstrate how to automatically obtain a wide array of labels for the synthesized objects with analytic label alignment. We comprehensively evaluate the effectiveness of Arti-PG toolbox on four representative object understanding tasks from both vision and robotic aspects, and the experiments suggest the superiority of our approach.

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

We provide comprehensive appendices for better understanding of our paper and offer more evidence to prove the effectiveness of our approach. The appendices are organized as follows: Appendix A-E first provide specific technical details and discussions about the implementation of Arti-PG. Then, more experimental results and analysis are presented in Appendix F and visualizations of our synthesized objects are shown in Appendix G. We further take the object category of '*Globe*' as an example to demonstrate how our approach is implemented in Python in Appendix H. Finally, we discuss about additional advantages behind our design, current limitations and further work in Appendix I.

## A  ARCHITECTURE AND OPERATING PRINCIPLES OF STRUCTURE PROGRAM

In Sec. 3.2, we introduced an example of a washing machine to show our design of the structure program and how to use it to describe the spatial structure of an articulated object. Here we provide another example to more clearly demonstrate the architecture and operating principles behind the program description of the object structure (in this case, the structure of a sliding window with two prismatic panels) step by step. This shows a better view of technical details such as the primitive composition of an object, how connectivity relationships work between primitives, and how advanced primitives are built upon elementary ones. For the rest of this section, we use two types of fonts, namely `monospaced` and *italic*, to indicate primitive instances and primitive class templates respectively.

**OBJECT STRUCTURE**: Let's start from the top row in Fig. 4, where the structure of the object is resolved into four components, `frame` and `window_1-3`, and all these windows are connected to `frame`. Particularly, `window_2` is in a fixed connection and `window_1`/`window_3` are in a prismatic connection. For fixed connection, we restrict the relative translations and rotations between `window_2` and `frame` to specific values. To implement the prismatic connection, we set the translation of `window_1`/`window_3` along the x-axis free within the length of `frame` and restrict the other relative translations and rotations between `window_1`/`window_3` and `frame`.

**Frame**: `Frame` is described with the primitive *rectangular_tube* and its corresponding parameters.

**Window_1-3**: `window_1-3` are instantiated from an advanced primitive of *window*, consisting of a window base and optional handles. The window base is described with a *concave_cuboid* elementary primitive and the handle is a *handle* advanced primitive, and the two components are connected with fixed connection. A discrete parameter is used to indicate the number of handles. Through the *window* advanced primitive, we can use *concave_cuboid* with different parameters and `handle_1-2` to describe `window_1-3`. Specifically, the number of handles is 0 for `window_2` and 1 for the rest. This also shows that the same primitive templates implemented with different parameters result in various structures.

**Handle_1-2**: `handle_1-2` are instantiated from an advanced primitive of *handle*, whose three components are all *cuboid* elementary primitives, and are instantiated into `handle_top`, `handle_middle` and `handle_bottom` in this case to construct both `handle_1-2`. The connection between `handle_top` and `handle_middle` is a fixed connection. Since `handle_middle` plays a role of a revolute joint, we connect `handle_bottom` with `handle_middle` by restricting their relative translations and rotations with the exception of the rotational freedom along the joint's axis of revolution. Together with these primitives and connectivity relationships between them, we get an advanced primitive template that describes a handle. By assigning specific parameters to the advanced primitive template, we are able to describe `handle_1-2`.

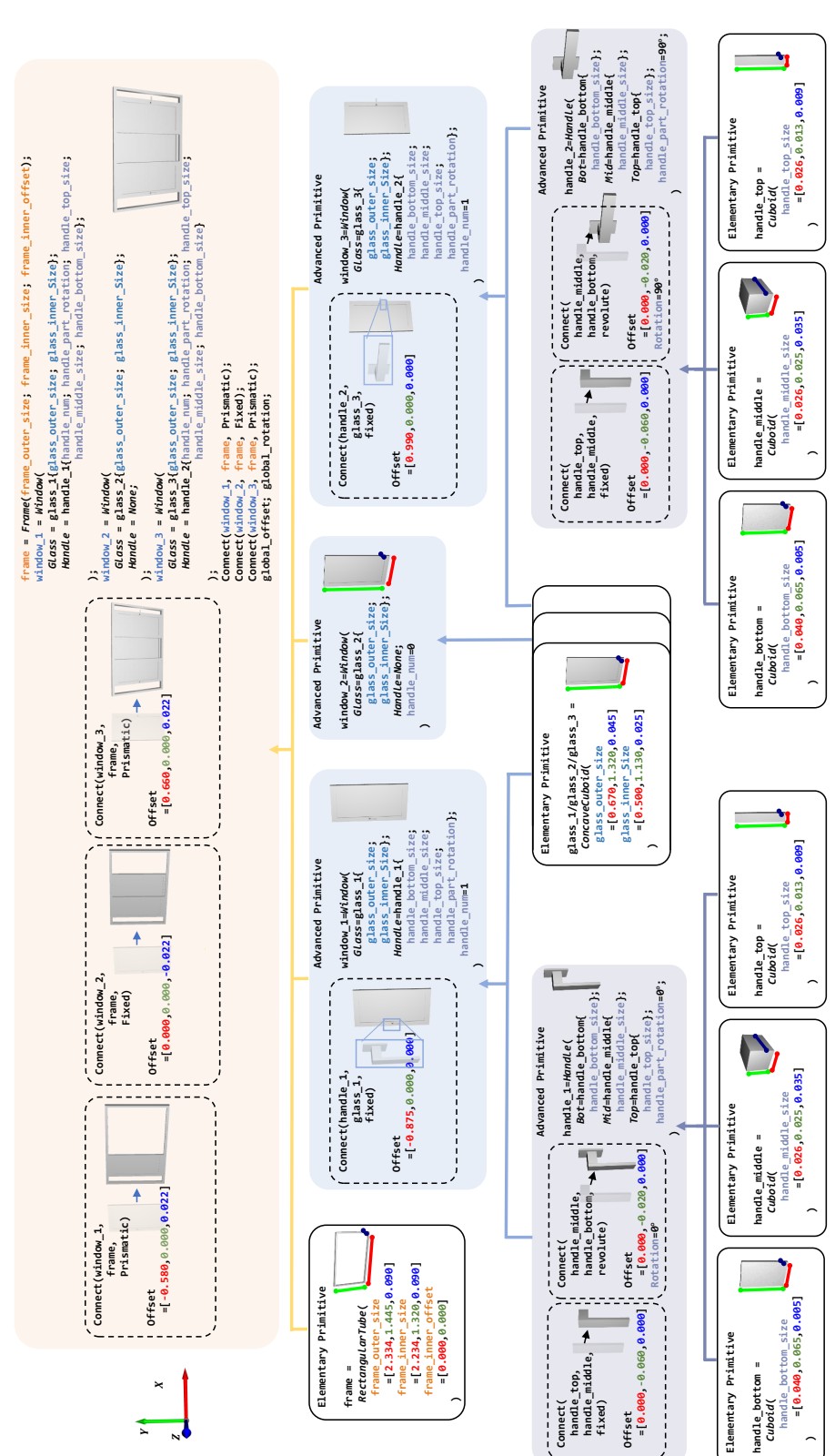

Figure 4: Architecture of the structure program of a window.

## B  POINTS ON GEOMETRIC PRIMITIVES

An important property of the geometric primitive is that each point on the primitive can be analytically described by mathematical functions, as discussed in footnote 2 of the main body. This property is crucial for the appearance representation and the label alignment process. In the footnote we give an example of a *sphere*, and here we further provide another example of a *cuboid* for better understanding. In this example, we assume that i) the center is at the origin and the orientations in terms of $L, H, W$ are aligned with the $x, y, z$-axes respectively in a *cuboid* primitive, and ii) $y_+$-axis points upward. Then, all the points on its top surface can be analytically described by $(\alpha L, \frac{1}{2}H, \beta W)$ with $\alpha, \beta \in [-\frac{1}{2}, \frac{1}{2}]$. A certain point on its top surface can also be designated by assigning specific value to $\alpha, \beta$, *e.g.* one of the corners can be represented by assigning both $\alpha$ and $\beta$ as $\frac{1}{2}$.

## C  EXAMPLE OF COMPLETING GEOMETRIC DETAILS

As mentioned in Sec. 3.5, there may be invisible points on the structure that are not covered by geometric details, and we deal with this issue by completing the geometric details separately for each elementary primitive according to its geometric property like translational and rotational symmetry. Here, we provide an example of such process on primitive cuboid.

We first assume that i) the center of the cuboid is at the origin and the orientations in terms of $L, H, W$ are aligned with the $x, y, z$-axes respectively, and ii) $\mathbb{U}$ is the set of all points on the primitive's surface and $\mathbb{V} \subset \mathbb{U}$ is the set of all visible points. For an invisible point $p = (x, y, z) \in \mathbb{U} - \mathbb{V}$, points that obey translational and rotational symmetry with $p$ compose a point set $\mathbb{S}$, written as

$$\mathbb{S} = \left\{ \begin{bmatrix} x \\ y \\ -z \end{bmatrix}, \begin{bmatrix} x \\ -y \\ z \end{bmatrix}, \begin{bmatrix} x \\ -y \\ -z \end{bmatrix}, \begin{bmatrix} -x \\ y \\ z \end{bmatrix}, \cdots \right\} \tag{3}$$

We select a point $q \in \mathbb{V}$ for each $p$ under following rules: i) $q$ is close enough to some point in $\mathbb{S}$

$$\exists s \in \mathbb{S}, \ \|q - s\|_2 < \epsilon \tag{4}$$

where $\epsilon$ is a threshold, and ii) adjacent $p$s should search for their corresponding $q$s in the same symmetric manner. Then, we can duplicate the appearance vector of $q$ to $p$. Finally, we apply a linear interpolation algorithm to fill the remaining holes if they exist and a filtering algorithm to make the appearance smoother. Fig. 5 gives a common case that results in invisible points in the contact surface of the lower cuboid, and we show one of the choices which adopts $s = \begin{bmatrix} x \\ -y \\ -z \end{bmatrix}$ to migrate geometric details to these points.

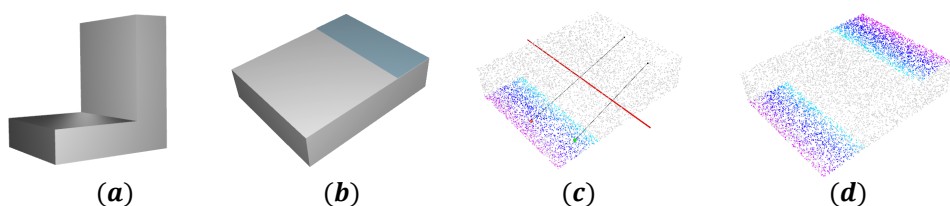

$(a)$       $(b)$       $(c)$       $(d)$

Figure 5: Example of completing geometric details for invisible points. **a.** A common case where two cuboids are stacked and the contact surface is invisible. **b.** The lower cuboid where the top right rectangular blue area indicates invisible area. **c.** One possible way to complete the geometric details on the invisible area is to migrate visible points *w.r.t.* axis symmetry along the red line. Black points on the top right are invisible points sampled on the cuboid surface. Green and red spheres show the searching area of corresponding points. Zoom in for a clear view. **d.** The result of completion.

## D    MORE EXAMPLES OF LABEL ALIGNMENT

Here we give more examples of automatically aligning labels onto synthesized objects, taking advantage of the analytic property.

**Part Semantics.**    The structure of an object is represented with a series of elementary geometric primitives in our program description. Since the elementary primitives typically serve as the foundational components in an object's hierarchy, we can obtain part semantics for each point by assigning a label to each elementary primitive (more specifically, all the points on it).

**Grasp Poses.**    Please refer to Fig. 6 for details.

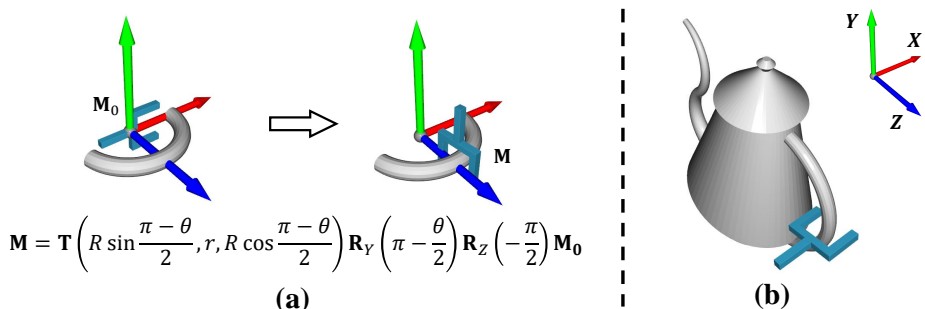

$$\mathbf{M} = \mathbf{T}\left(R\sin\frac{\pi-\theta}{2}, r, R\cos\frac{\pi-\theta}{2}\right)\mathbf{R}_Y\left(\pi-\frac{\theta}{2}\right)\mathbf{R}_Z\left(-\frac{\pi}{2}\right)\mathbf{M}_0$$

**(a)**                                                                        **(b)**

Figure 6: Illustration of analytically aligning grasp poses. (a) We first label a grasp pose of the primitive, *i.e.* a torus segment in this example, by transforming the gripper from its initial pose ($\mathbf{M}_0$) to a proper grasp pose ($\mathbf{M}$) using the mathematical expression below. Here the major radius $R$ is the distance from the center of the tube to the center of the torus, the minor radius $r$ is the radius of the tube, and $\theta$ is the segment angle. $\mathbf{R}_*$ denotes the transformation matrix for rotation around axis *, $\mathbf{T}$ denotes the transformation matrix for translation. (b) With the grasp pose aligned to the torus segment, a synthesized kettle is automatically labelled with this affordance when the torus segment is used in the structure as a handle.

## E    DETAILS OF STRUCTURE PROGRAM ANNOTATION SYSTEM

We have elaborately designed a user-friendly annotation system to efficiently and effectively obtain the structure program of a real object. It is a web-based system, allowing users to easily access it through a browser. The system is designed as a one-way question-answering workflow, where users are tasked to determine the primitives and specify their parameters for a given object. During annotation, real-time renderings of the structural program as well as the target object itself are shown on the web page in a synchronous way for reference. We also show a mixed view of the two renderings for better comparison. We provide a video demonstration of the system in **anno_system_videos/system_demo.mp4** of supplementary material. Some of our codes are borrowed from PartNet Anno System (daerduoCarey  (Kaichun Mo)).

In practice, we invite first-year undergraduate students to assist us in the annotation process, since it just requires high-school level math skills. For reference, the average annotation time for an object is about 6 mins. To demonstrate the annotation process in detail, we provide a video of annotation footage featuring three volunteers in **anno_system_videos/anno_footage.mp4** in supplementary material. This shows that the system is user-friendly and efficient in obtaining structure programs.

## F    MORE DETAILS ON EXPERIMENTS

**Vision Experiment Settings.**    Here we provide more details for vision tasks settings. In Tab. 5, we give the detailed statistics of our dataset in terms of train and test set sizes. For part segmentation, besides PointTransformer (Zhao et al. (2021)) as the baseline mentioned in the main body, we

further introduce the classical PointNet++ (Qi et al. (2017a)) as another baseline to further demonstrate our approach's effectiveness. PointNet++ is an efficient and effective network which serves as the backbone of many 3D frameworks. For part pose estimation, we follow GAPartNet (Geng et al. (2023)) for data preparation. Specifically, we render RGB-D images of articulated objects in SAPIEN simulator (Xiang et al. (2020)) with annotations, variate collected data by using random camera poses and joint poses and finally gather 20000 points as input. The position and orientation of parts are defined in the Normalized Part Coordinate Space (NPCS). Specifically, each detectable part is reduced to a standard orientation and normalized within a unit ball. We use batch sizes from 16 to 64 for different tasks, depending on the default settings of baseline models. We use Adam optimizer with learning rate = 0.001 and weight decay = 0.0001 to optimize the network parameters.

Table 5: Detailed statistics of the data split on vision tasks.

| Split | Bottle | Box | Bucket | Display | Door | Eyeglasses | Globe | Kettle |
|---|---|---|---|---|---|---|---|---|
| Train | 64 | 18 | 18 | 50 | 24 | 43 | 40 | 18 |
| Test | 400 | 10 | 18 | 904 | 12 | 22 | 20 | 10 |
| | KitchenPot | Laptop | Lighter | Microwave | Pen | Pliers | Fridge | Safe |
| Train | 15 | 48 | 18 | 6 | 32 | 10 | 30 | 20 |
| Test | 10 | 405 | 10 | 10 | 16 | 14 | 14 | 10 |
| | Scissors | Stapler | Switch | TrashCan | USB | Washing | Window | |
| Train | 32 | 13 | 47 | 37 | 20 | 7 | 35 | |
| Test | 15 | 10 | 23 | 19 | 31 | 10 | 18 | |

**More Vision Task Results.** We provide part segmentation and point cloud completion results for each object category in detail in Tab. 6, as well as a new part segmentation baseline PointNet++. Since part pose estimation is not a category-level task, we do not provide per category results of this task. For both tasks and all the baselines, our approach is able to provide significant improvement across all the object categories. Further, our approach surpasses the data augmentation approach PointWOLF in the segmentation task for almost all categories, especially for categories with more delicate structures, *e.g.* Pliers and USB. This can be attributed to Arti-PG's capability of synthesizing structures with a wide range of variety while ensuring their validity, whereas PointWOLF, augmenting based on random local transformations that may potentially harm the structural integrity of such delicate objects, begins to show negative impacts on the performance. These results provide more comprehensive evidence of the superiority of our approach.

**Manipulation Experiment Settings.** As shown in Tab. 7, we conduct our experiment on 15 representative categories of objects. We would like our evaluation to reflect the ability to understand articulated object structures and detect affordances on articulated objects rather than delicate trajectory planning. Hence we have removed objects that are either too small (*e.g.* Pen, USB) or do not make sense for a single-gripper to manipulate (*e.g.* Bottle, Scissors). This practice follows the baseline (Mo et al. (2021)).

We adapt the SAPIEN (Xiang et al. (2020)) simulator as the interaction environment for manipulation tasks. For each interaction simulation, we initially place an object in the SAPIEN simulator at the center of the scene. The joint pose of the object has a $50\%$ chance of being at the closed state (*e.g.* a closed door) and a $50\%$ chance of being at the open state with random motion (e.g. a half open closet). The whole scene is observed through an RGB-D camera with known intrinsic parameters, which stares at the center of the object and is positioned at the upper hemisphere with a random azimuth [0°,360°) and a random altitude [30°,60°]. A Franka Panda Flying gripper with 2 fingers is used to interact with the object.

For Where2Act (Mo et al. (2021)) and Where2Explore (Ning et al. (2024)), the per-pixel action likelihoods and action proposals are acquired by the networks. We select the pixel with the maximum action likelihood as the target and adopt the orientation and movement direction of the gripper given by the action proposal at this point. GAPartNet (Geng et al. (2023)) detects actionable parts with their poses on objects. The gripper orientation and movement direction are acquired based on the

Table 6: Per category experimental results on part segmentation and completion. *Impr.* denotes the improvement of Arti-PG over the baseline in absolute value.

| Network & Metric | Method | Bot | Box | Buc | Dis | Door | Eye | Glb | Ket | Pot | Ltp | Lit | Wav |
|---|---|---|---|---|---|---|---|---|---|---|---|---|---|
| Point-Transformer mAcc(%)↑ | × | 95.4 | 95.4 | 96.3 | 93.9 | 77.9 | 96.5 | 95.9 | 89.7 | 90.3 | 96.8 | 92.3 | 82.8 |
| | Arti-PG | **96.6** | **97.2** | **98.5** | **96.4** | **78.1** | **97.1** | **96.9** | **93.9** | **95.8** | **97.1** | **93.8** | **90.3** |
| | *Impr.* | 1.2 | 1.8 | 2.2 | 2.5 | 0.2 | 0.6 | 1.0 | 4.2 | 5.5 | 0.3 | 1.5 | 7.4 |
| | PointWOLF | 96.8 | 96.2 | 92.6 | 95.3 | 77.5 | 96.3 | 95.5 | 90.5 | 91.1 | 96.8 | 92.7 | 87.2 |
| | | Pen | Pli | Fri | Safe | Sci | Stp | Swi | Can | USB | WM | Win | AVG |
| | × | 85.7 | 74.0 | 94.1 | 92.8 | 90.5 | 79.9 | 84.5 | 92.2 | 82.6 | 91.6 | 87.2 | 89.5 |
| | Arti-PG | **87.6** | **75.2** | **94.3** | **94.6** | **90.6** | **82.8** | **85.0** | **92.7** | **82.8** | **92.1** | **91.6** | **91.3** |
| | *Impr.* | 1.9 | 1.2 | 0.2 | 1.8 | 0.1 | 2.9 | 0.5 | 0.5 | 0.2 | 0.5 | 4.4 | 1.8 |
| | PointWOLF | 86.5 | 71.4 | 94.1 | 94.3 | 90.5 | 77.5 | 89.5 | 92.1 | 80.1 | 91.2 | 87.6 | 89.7 |
| | | Bot | Box | Buc | Dis | Door | Eye | Glb | Ket | Pot | Ltp | Lit | Wav |
| Point-Transformer mIoU(%)↑ | × | 75.1 | 93.7 | 48.5 | 81.9 | 52.1 | 92.9 | 85.5 | 88.7 | 92.8 | 87.7 | 72.5 | 74.5 |
| | Arti-PG | **82.7** | **96.4** | **49.8** | **84.0** | 56.0 | **94.1** | **94.2** | **93.5** | **97.6** | **88.8** | **84.1** | **81.1** |
| | *Impr.* | 7.6 | 2.7 | 1.3 | 2.1 | 3.9 | 1.2 | 8.7 | 4.8 | 4.7 | 1.0 | 11.6 | 6.6 |
| | PointWOLF | 82.2 | 94.7 | 49.3 | 82.5 | **59.6** | 93.9 | 87.9 | 89.5 | 93.6 | 88.4 | 73.1 | 74.9 |
| | | Pen | Pli | Fri | Safe | Sci | Stp | Swi | Can | USB | WM | Win | AVG |
| | × | 65.9 | 75.6 | 61.3 | 86.8 | 56.5 | 74.3 | 71.0 | 72.7 | 87.6 | 48.2 | 68.2 | 74.5 |
| | Arti-PG | **66.6** | **88.5** | **64.9** | **89.0** | **61.5** | **83.5** | 71.8 | **82.9** | **88.6** | **53.3** | **73.0** | **79.4** |
| | *Impr.* | 0.7 | 12.9 | 3.6 | 2.2 | 5.0 | 9.2 | 0.8 | 10.2 | 1.0 | 5.1 | 0.8 | 4.9 |
| | PointWOLF | 65.7 | 82.8 | 62.6 | 87.5 | 60.8 | 70.7 | **72.4** | 71.6 | 81.9 | 47.9 | 70.4 | 75.8 |
| | | Bot | Box | Buc | Dis | Door | Eye | Glb | Ket | Pot | Ltp | Lit | Wav |
| Pointnet++ mAcc(%)↑ | × | 95.5 | 93.7 | 98.1 | 91.0 | 81.0 | 97.4 | 88.0 | 87.5 | 92.1 | 96.5 | 91.6 | 86.5 |
| | Arti-PG | **95.6** | **95.8** | **98.6** | **94.6** | **82.4** | 97.5 | **95.9** | **93.2** | **96.0** | **97.2** | **93.8** | **89.8** |
| | *Impr.* | 0.1 | 2.1 | 0.5 | 3.6 | 1.4 | 0.1 | 7.9 | 5.7 | 3.9 | 0.7 | 2.2 | 3.3 |
| | PointWOLF | 95.5 | 94.3 | 98.4 | 92.5 | 81.1 | **97.9** | 89.2 | 92.8 | 92.8 | 97.0 | 91.5 | 87.9 |
| | | Pen | Pli | Fri | Safe | Sci | Stp | Swi | Can | USB | WM | Win | AVG |
| | × | 88.4 | 68.9 | 92.9 | 90.4 | 88.4 | 78.8 | 84.5 | 91.2 | 80.8 | 91.4 | 82.2 | 88.6 |
| | Arti-PG | **89.4** | **69.5** | **93.1** | **91.7** | 90.0 | 79.5 | **88.9** | **92.3** | **82.3** | **92.6** | **88.8** | **90.8** |
| | *Impr.* | 1.0 | 0.6 | 0.2 | 1.3 | 1.6 | 0.7 | 4.5 | 1.1 | 1.5 | 1.2 | 6.6 | 2.0 |
| | PointWOLF | 88.5 | 68.5 | 93.0 | 90.1 | **90.3** | 80.2 | 85.4 | 91.2 | 80.6 | 91.6 | 83.2 | 89.1 |
| | | Bot | Box | Buc | Dis | Door | Eye | Glb | Ket | Pot | Ltp | Lit | Wav |
| Pointnet++ mIoU(%)↑ | × | 71.9 | 83.7 | 54.8 | 61.1 | 51.6 | 93.8 | 77.0 | 59.5 | 84.4 | 83.0 | 60.0 | 67.3 |
| | Arti-PG | **73.1** | **89.7** | **57.2** | **77.3** | **56.3** | 94.3 | **91.2** | **78.8** | **90.0** | **83.4** | **67.4** | **74.5** |
| | *Impr.* | 1.3 | 6.0 | 2.4 | 16.2 | 4.7 | 0.5 | 14.2 | 19.3 | 5.6 | 0.4 | 7.4 | 7.2 |
| | PointWOLF | 73.0 | 86.1 | 55.3 | 64.4 | 51.4 | **95.1** | 78.0 | 60.8 | 84.7 | 83.3 | 59.6 | 70.3 |
| | | Pen | Pli | Fri | Safe | Sci | Stp | Swi | Can | USB | WM | Win | AVG |
| | × | 69.4 | 60.7 | 58.7 | 63.5 | 62.1 | 61.7 | 47.1 | 69.7 | 73.2 | 55.7 | 62.1 | 66.6 |
| | Arti-PG | **71.6** | **67.5** | **60.7** | **66.5** | **66.7** | **63.4** | **59.7** | **78.4** | **76.4** | **66.6** | **74.9** | **73.3** |
| | *Impr.* | 2.2 | 6.8 | 2.0 | 3.0 | 4.6 | 1.7 | 12.6 | 8.7 | 3.2 | 10.9 | 12.8 | 6.7 |
| | PointWOLF | 68.9 | 59.3 | 60.2 | 62.8 | 65.3 | 63.2 | 51.2 | 70.0 | 69.0 | 57.7 | 64.8 | 67.5 |
| | | Bot | Box | Buc | Dis | Door | Eye | Glb | Ket | Pot | Ltp | Lit | Wav |
| SnowflakeNet CD(×10⁻⁴)↓ | × | 9.7 | 14.6 | 14.4 | 9.4 | 8.6 | 5.1 | 18.2 | 19.4 | 17.6 | 9.4 | 8.6 | 15.8 |
| | Arti-PG | **9.6** | **14.0** | **13.0** | **9.3** | **8.5** | 5.1 | **17.0** | **19.0** | **16.4** | **7.1** | **7.2** | **13.3** |
| | *Impr.* | 0.1 | 0.6 | 1.4 | 0.1 | 0.1 | 0.0 | 1.2 | 0.4 | 1.2 | 2.3 | 1.4 | 2.5 |
| | | Pen | Pli | Fri | Safe | Sci | Stp | Swi | Can | USB | WM | Win | AVG |
| | × | 4.7 | 6.5 | 8.9 | 15.2 | 5.0 | 9.6 | 13.6 | 12.7 | 8.9 | 16.2 | 6.9 | 11.3 |
| | Arti-PG | **4.7** | **5.3** | **8.8** | **12.2** | **4.6** | **8.4** | 13.6 | **12.0** | **8.0** | **16.0** | **5.3** | **10.4** |
| | *Impr.* | 0.0 | 1.2 | 0.1 | 3.0 | 0.4 | 1.2 | 0.0 | 0.7 | 0.9 | 0.2 | 1.6 | 0.9 |

part pose, namely we turn the gripper in an orientation suitable for grasping and move the gripper toward/away from the target part.

Tab. 8 lists specific manipulation tasks on our objects. The tasks can be generally categorized into pushing and pulling. Specifically, for pushing tasks, a closed gripper is initially placed 0.05m away from the target along the movement direction, then moves forward with a longer distance in order to push the target. For pulling tasks, an open gripper is placed 0.05m away from the target along the movement direction, then moves forward to the target with 0.045m and closes itself to grasp the target. The gripper subsequently moves back to the start point to pull the target.

**Detailed Manipulation Results.** We provide manipulation results for each object category in detail in Tab. 9. Further, video demonstrations for manipulation in both simulation and real world environment are provided in **experiment_videos** in supplementary material.

Table 7: Detailed statistics of the data split on manipulation tasks.

| Train Cats | Box | Door | Faucet | Kettle | Microwave |
|---|---|---|---|---|---|
| Train | 20 | 23 | 65 | 22 | 9 |
| Test | 8 | 12 | 19 | 7 | 3 |
| | Fridge | Storage | Switch | TrashCan | Window |
| | 32 | 270 | 53 | 52 | 40 |
| | 11 | 75 | 17 | 17 | 18 |
| Test Cats | Bucket | KitchenPot | Safe | Table | Washing |
| Test | 36 | 23 | 29 | 95 | 16 |

Table 8: List of specific tasks in manipulation. The tasks can be generally categoried into pushing and pulling.

| Category | Tasks |
|---|---|
| Box | Push/Pull Lid |
| Bucket | Push/Pull Handle |
| Door | Push Door; Push/Pull Door via Handle |
| Faucet | Push/Pull Switch |
| Fridge | Push Door; Push/Pull Door via Handle |
| Kettle | Push/Pull Handle |
| KitchenPot | Push/Pull Handle; Pull Lid |
| Microwave | Push Door; Push/Pull Door via Handle |
| Safe | Push Door; Push/Pull Door via Handle |
| StorageFurniture | Push Door; Push/Pull Door via Handle; Push/Pull Drawer via Handle |
| Switch | Push/Pull Switch |
| Table | Push Door; Push/Pull Door via Handle; Push/Pull Drawer via Handle |
| TrashCan | Push/Pull Lid |
| WashingMachine | Push Door; Push/Pull Door via Handle; Push Lid |
| Window | Push Window; Push/Pull Window via Handle |

**Amount of Available Data.** To fully demonstrate the potential of our approach in the data scarcity scenario, we further conduct ablation studies by gradually reducing the number of real objects in the training set from 100% to 1% (at least 1 object in each category for training). Results in Fig. 7 suggest that more benefits can be yielded by Arti-PG on a smaller training set, *i.e.* the data scarcity issue is more prominent.

Table 9: Per category experimental results on manipulation. All values are percentage sample success rate. *Impr.* denotes the improvement of Arti-PG over the baseline in absolute ssr.

| Network | Task | Method | Box | Buc | Door | Fau | Fri | Ket | Mic | Pot | Safe | Sto | Swi | Tab | Tra | Was | Win | AVG |
|---|---|---|---|---|---|---|---|---|---|---|---|---|---|---|---|---|---|---|
| W2A | push | × | 25.8 | 8.2 | 34.1 | 27.9 | 32.2 | 23.7 | 35.8 | 6.2 | 9.8 | 32.9 | 28.0 | 21.0 | 19.0 | 13.0 | 15.9 | 21.4 |
| | | Arti-PG | 32.8 | 12.3 | 38.0 | 29.1 | 37.3 | 29.1 | 40.4 | 7.1 | 13.5 | 36.1 | 31.5 | 30.6 | 21.2 | 18.1 | 20.9 | 26.4 |
| | | Impr. | 7.0 | 4.1 | 3.9 | 1.2 | 5.1 | 5.4 | 4.6 | 0.9 | 3.7 | 3.2 | 3.5 | 9.6 | 2.2 | 5.1 | 5.0 | 5.0 |
| | pull | × | 3.4 | 6.1 | 4.7 | 5.5 | 5.2 | 3.0 | 6.0 | 3.6 | 5.6 | 10.7 | 9.1 | 10.5 | 5.5 | 5.9 | 3.3 | 7.6 |
| | | Arti-PG | 4.5 | 7.9 | 6.5 | 11.1 | 5.7 | 5.0 | 8.0 | 5.0 | 5.7 | 11.7 | 9.8 | 12.6 | 6.3 | 7.4 | 4.0 | 9.2 |
| | | Impr. | 1.1 | 1.8 | 1.8 | 5.6 | 0.5 | 2.0 | 2.0 | 1.4 | 0.1 | 1.0 | 0.7 | 2.1 | 0.8 | 1.6 | 0.7 | 1.6 |
| W2E | push | × | 38.0 | 15.2 | 39.5 | 31.9 | 46.8 | 21.5 | 36.8 | 10.8 | 13.9 | 37.9 | 23.8 | 24.2 | 30.3 | 16.9 | 17.0 | 25.9 |
| | | Arti-PG | 40.5 | 20.4 | 45.0 | 34.0 | 47.2 | 28.3 | 44.1 | 15.7 | 16.0 | 43.9 | 27.7 | 37.2 | 40.0 | 20.5 | 24.0 | 32.8 |
| | | Impr. | 2.5 | 5.2 | 5.5 | 2.1 | 0.4 | 6.8 | 7.3 | 4.9 | 2.1 | 6.0 | 3.9 | 13.0 | 9.7 | 3.6 | 7.0 | 6.9 |
| | pull | × | 6.6 | 9.4 | 8.8 | 7.0 | 12.5 | 5.3 | 7.5 | 6.2 | 9.2 | 10.9 | 11.8 | 10.7 | 11.8 | 4.5 | 2.5 | 9.3 |
| | | Arti-PG | 8.5 | 15.9 | 13.3 | 11.3 | 14.5 | 8.9 | 12.9 | 12.7 | 11.3 | 12.2 | 14.3 | 10.9 | 16.1 | 8.7 | 3.6 | 11.9 |
| | | Impr. | 1.9 | 6.5 | 4.5 | 4.3 | 2.0 | 3.6 | 5.4 | 6.5 | 2.1 | 1.3 | 2.5 | 0.2 | 4.3 | 4.2 | 1.1 | 2.6 |
| GA | push | × | 41.7 | 25.5 | 47.9 | 18.0 | 45.1 | 34.4 | 37.1 | 19.0 | 9.3 | 38.8 | 19.1 | 24.8 | 25.0 | 14.8 | 15.6 | 26.6 |
| | | Arti-PG | 43.2 | 35.0 | 52.1 | 31.0 | 52.8 | 39.6 | 40.4 | 23.1 | 13.7 | 41.2 | 31.7 | 34.9 | 31.8 | 18.3 | 24.3 | 33.5 |
| | | Impr. | 1.5 | 9.5 | 4.2 | 13.0 | 7.7 | 5.2 | 3.3 | 4.1 | 4.4 | 2.4 | 12.6 | 10.1 | 6.8 | 3.5 | 8.7 | 6.9 |
| | pull | × | 10.1 | 17.0 | 12.3 | 6.7 | 13.9 | 10.6 | 10.9 | 6.8 | 9.3 | 16.6 | 11.1 | 16.3 | 9.0 | 7.9 | 2.6 | 12.9 |
| | | Arti-PG | 11.3 | 26.4 | 14.0 | 7.0 | 16.5 | 15.1 | 16.0 | 14.1 | 10.4 | 19.3 | 12.1 | 20.8 | 10.3 | 10.1 | 4.7 | 16.5 |
| | | Impr. | 1.2 | 9.4 | 1.7 | 0.3 | 2.6 | 4.5 | 5.1 | 7.3 | 1.1 | 2.7 | 1.0 | 4.5 | 1.3 | 2.2 | 2.1 | 3.6 |

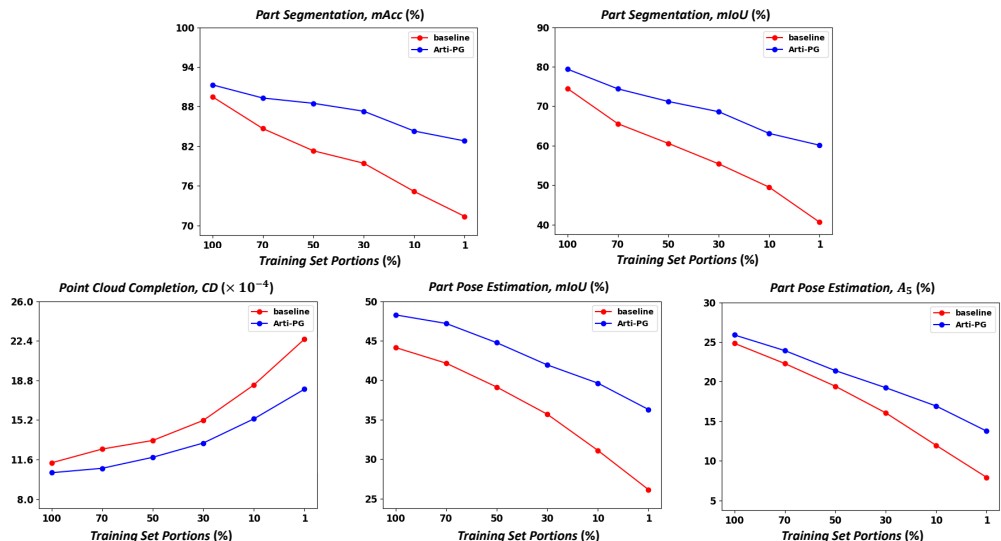

Figure 7: Performance of both baseline and Arti-PG on various tasks with respect to changes in the portions of available data. The results are reported on average across all categories.

**Results with Sufficient Training Data.** Although we focus on solving the data scarcity issue, we would like to demonstrate that our approach also works in scenarios with sufficient training data. We pick *Bottle*, *Display* and *Laptop* where enough data are available. We re-split the training and test sets of these categories to construct two settings for this experiment: sufficient training data and scarce training data. The data split is reported in Tab. 10, where the same 100 data are used for the test. We evaluate our approach in both settings across part segmentation and point cloud completion tasks with mean accuracy (mAcc), mean IoU (mIoU) and Chamfer Distance (CD) as metrics. Tab. 11 demonstrates the results, which suggest that our approach is still effective with a large number of training data.

Table 10: Data split for sufficient training data and scarce training data scenario.

| Setting | Bottle | Display | Laptop |
|---------|--------|---------|--------|
| Sufficient Train Data | 364 | 854 | 353 |
| Scarce Train Data | 64 | 50 | 48 |
| Test Data | 100 | 100 | 100 |

Table 11: Experimental results of part segmentation and point cloud completion on two settings: sufficient training data and scarce training data.

| Task | Metric | Method | Sufficient Train Data | | | Scarce Train Data | | |
|------|--------|--------|--------|---------|--------|--------|---------|--------|
| | | | Bottle | Display | Laptop | Bottle | Display | Laptop |
| Segmentation | **mAcc**(%) ↑ | × | 95.8 | 96.0 | 97.1 | 94.3 | 93.8 | 95.4 |
| | | Arti-PG | **96.9** | **96.5** | **97.4** | **95.5** | **95.5** | **96.5** |
| | | *Impr.* | 1.1 | 0.5 | 0.3 | 1.2 | 1.7 | 1.1 |
| | **mIoU**(%) ↑ | × | 80.8 | 88.5 | 84.1 | 70.8 | 75.9 | 83.4 |
| | | Arti-PG | **83.8** | **88.8** | **85.6** | **80.6** | **80.6** | **84.9** |
| | | *Impr.* | 3.0 | 0.3 | 1.5 | 9.8 | 4.7 | 1.5 |
| Completion | CD($\times 10^{-4}$cm)↓ | × | 7.369 | 8.942 | 7.443 | 10.079 | 9.671 | 9.289 |
| | | Arti-PG | **6.187** | **8.752** | **6.637** | **9.012** | **9.312** | **8.117** |
| | | *Impr.* | 1.182 | 0.190 | 0.806 | 1.067 | 0.359 | 1.172 |

## G    VISUALIZATIONS OF SYNTHESIZED OBJECTS

Here, we provide substantial illustrations of synthesized objects from 26 categories in Fig. 8, 9, 10, 11 and 12. This demonstrates that our approach is capable of synthesizing high-quality 3D articulated objects with considerable diversity in both structure and appearance.

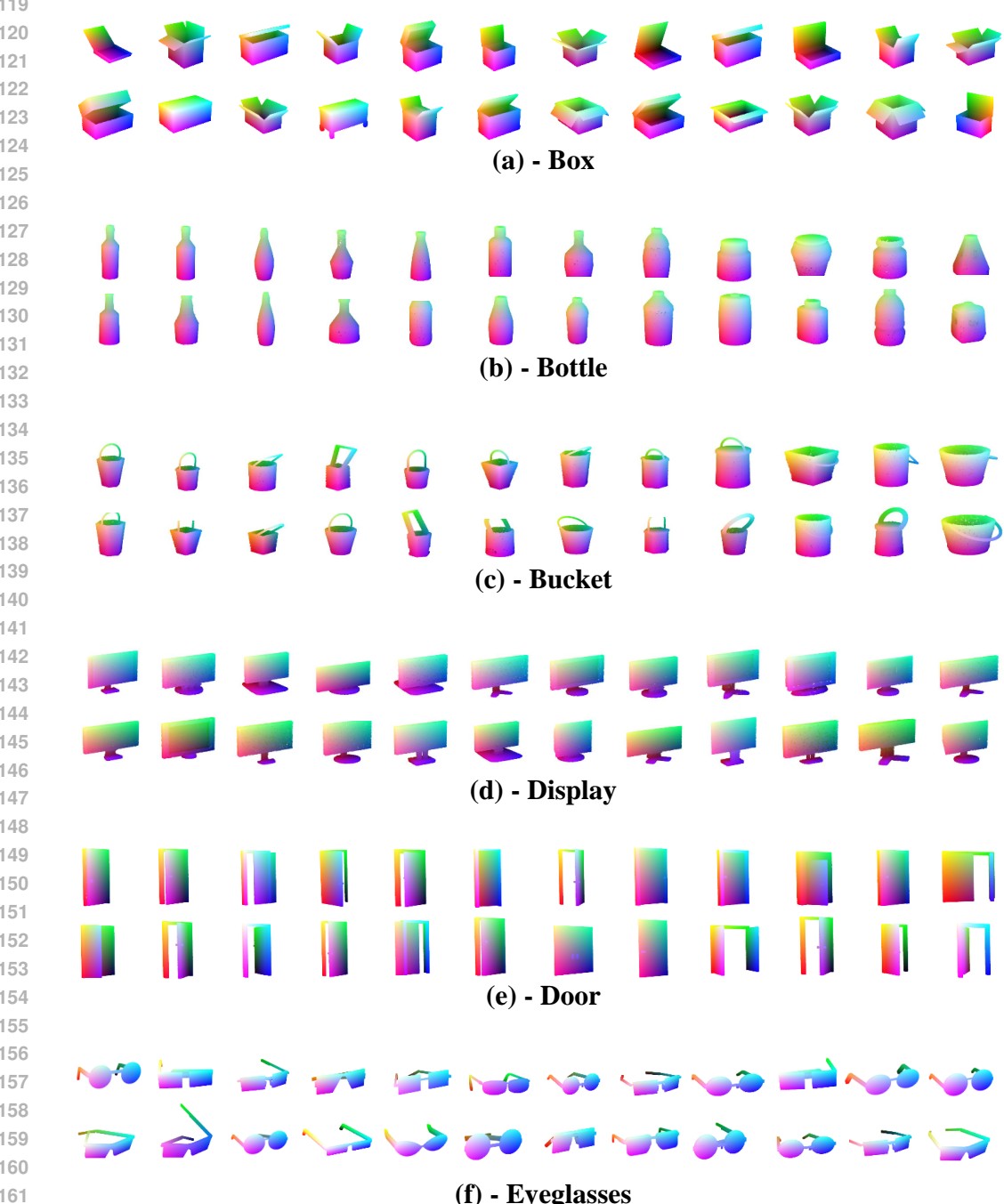

**(a) - Box**

**(b) - Bottle**

**(c) - Bucket**

**(d) - Display**

**(e) - Door**

**(f) - Eyeglasses**

Figure 8: Various categories of objects synthesized by Arti-PG. Part I.

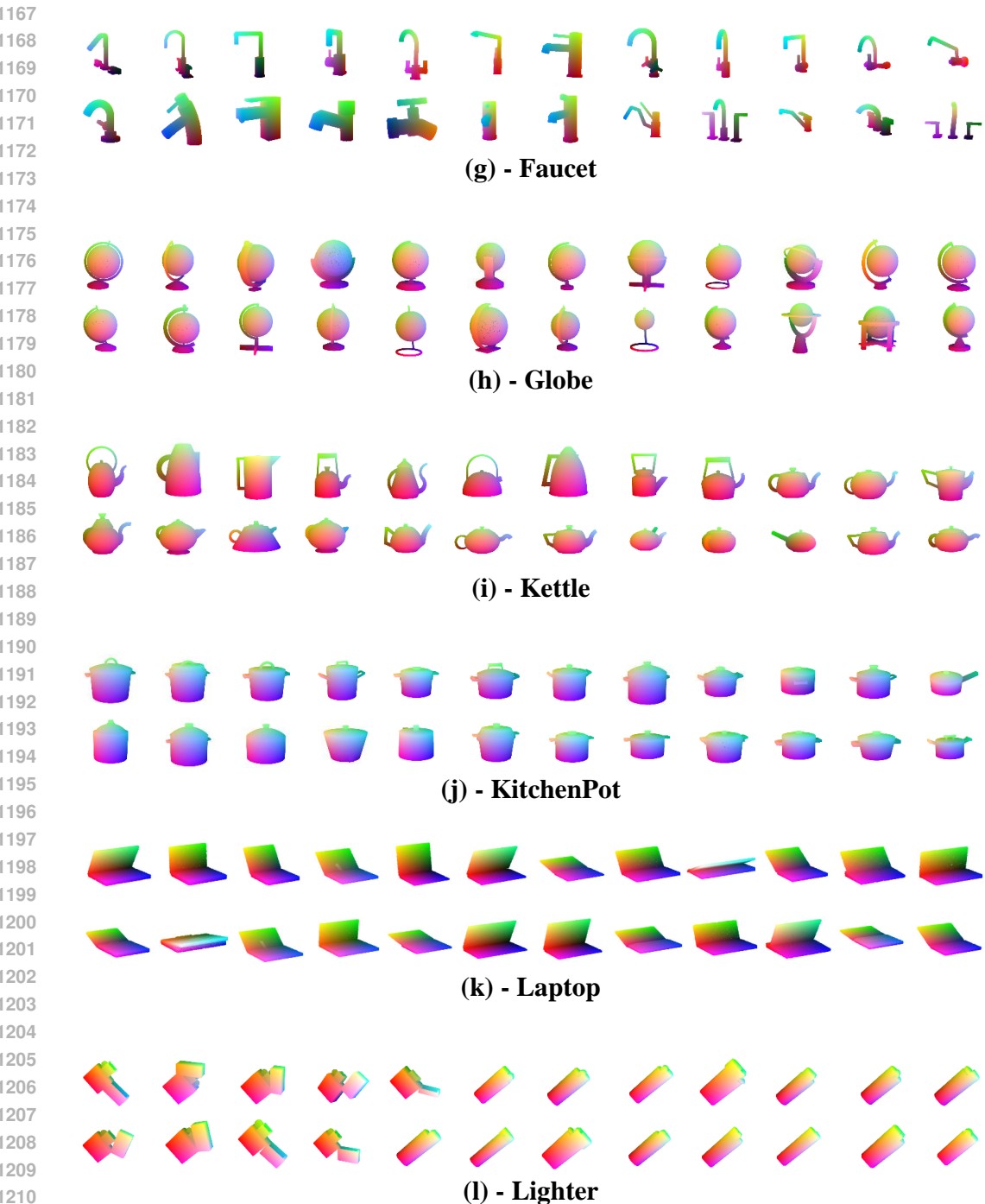

(g) - Faucet

(h) - Globe

(i) - Kettle

(j) - KitchenPot

(k) - Laptop

(l) - Lighter

Figure 9: Various categories of objects synthesized by Arti-PG. Part II.

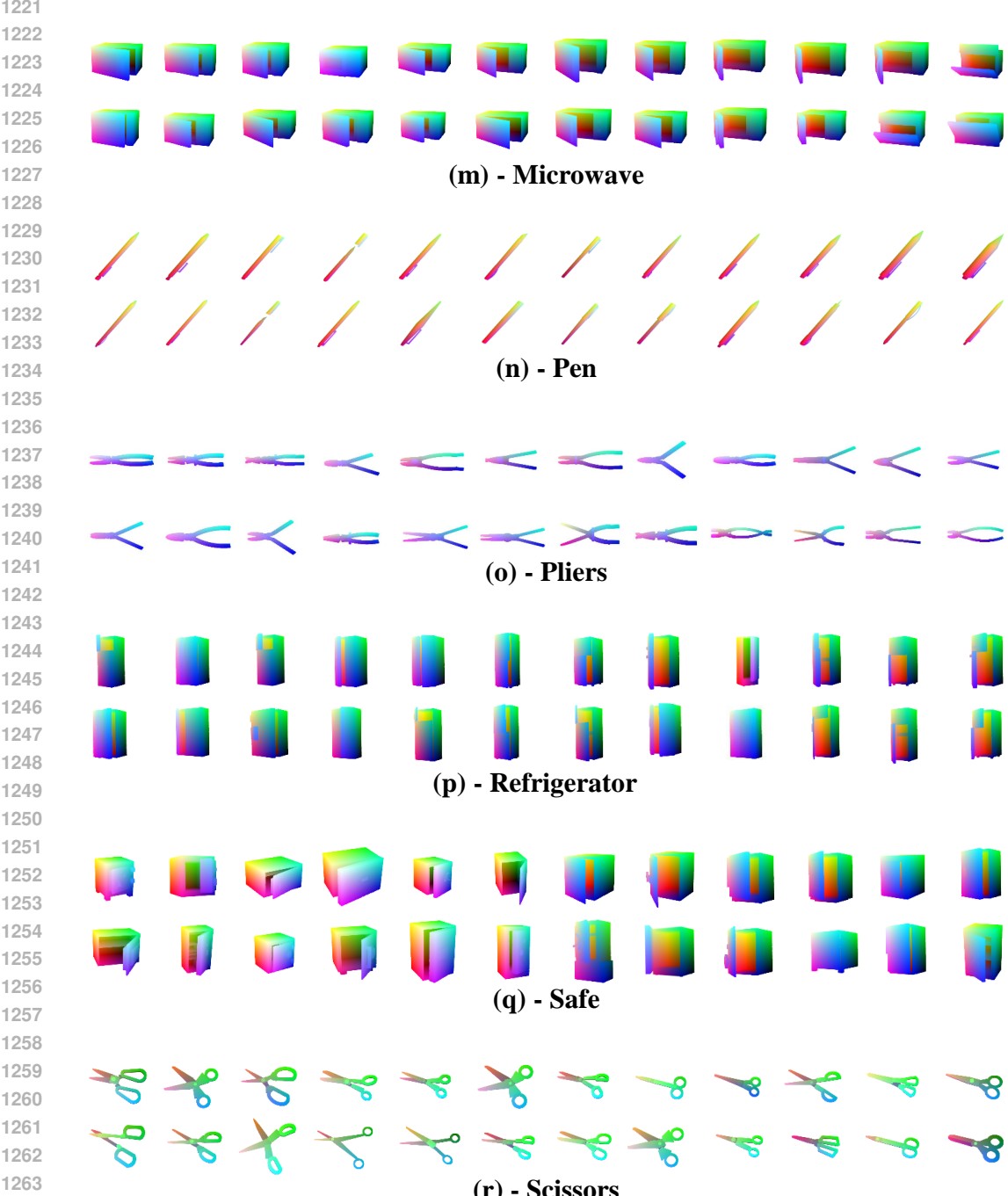

**(m) - Microwave**

**(n) - Pen**

**(o) - Pliers**

**(p) - Refrigerator**

**(q) - Safe**

**(r) - Scissors**

Figure 10: Various categories of objects synthesized by Arti-PG. Part III.

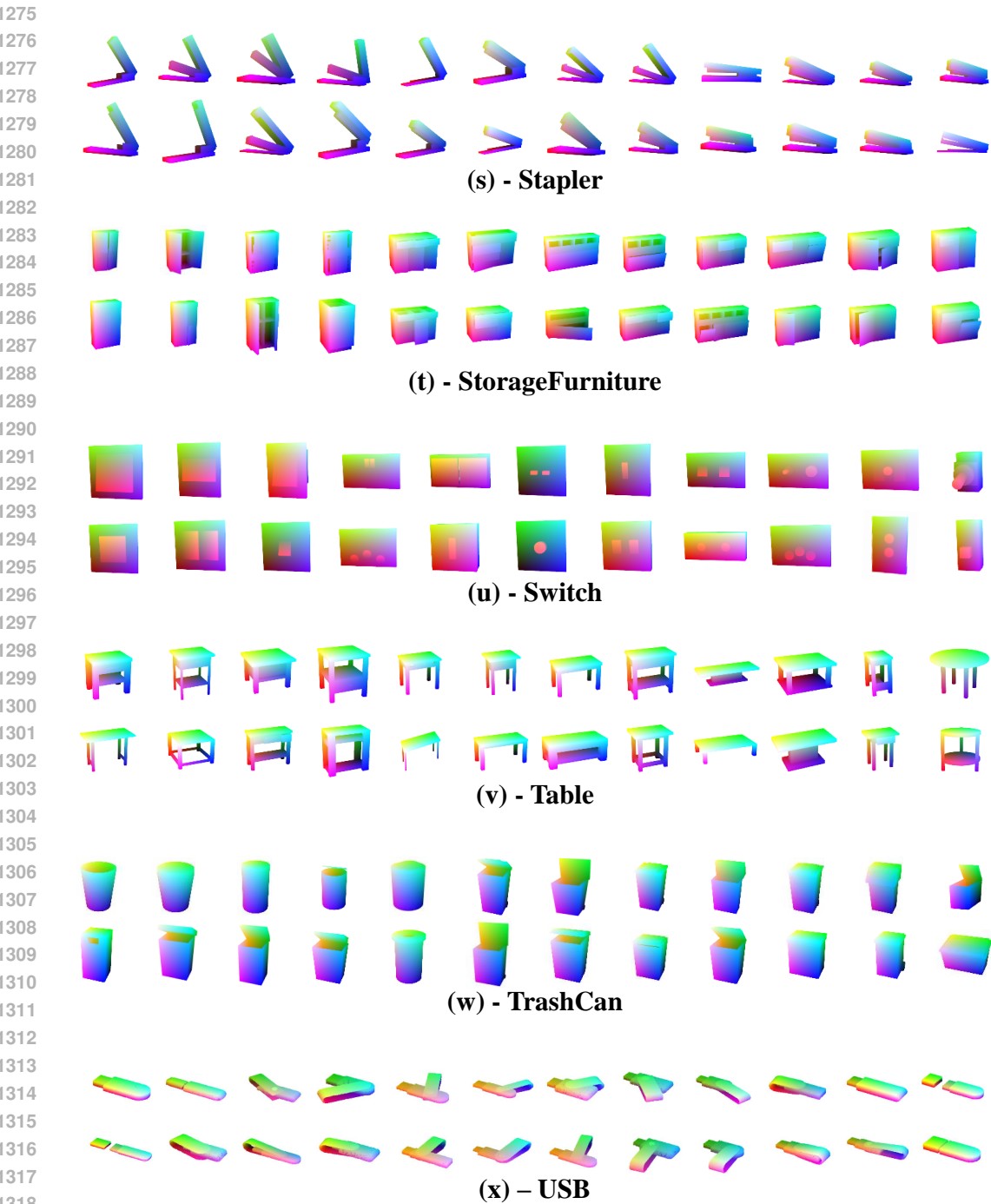

**(s) - Stapler**

**(t) - StorageFurniture**

**(u) - Switch**

**(v) - Table**

**(w) - TrashCan**

**(x) – USB**

Figure 11: Various categories of objects synthesized by Arti-PG. Part IV.

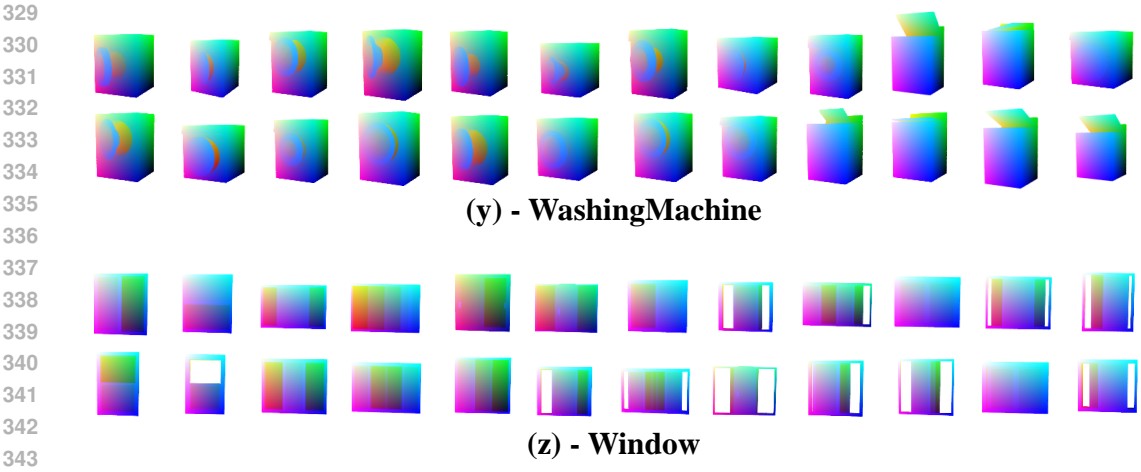

**Figure 12**: Various categories of objects synthesized by Arti-PG. Part V.

## H IMPLEMENTATION OF STRUCTURE PROGRAMS IN PYTHON

In this section, we show the implementation of the structure programs in Python and provide detailed explanations, taking '*Globe*' as an example. For simplicity, we omit ancillary codes like "converting `List` type to `numpy.ndarray` type". Our codes for all object categories will be made publicly available.

### H.1 ELEMENTARY PRIMITIVES

**Base Class.** First, we implement the base class for elementary primitives. It mainly contains the offset and rotation of an elementary primitive. The elementary primitive can be further moved in 3D space through functions like `translate` and `rotate`.

```python
class Elementary_Primitive:
    def __init__(
        self,
        offset=[0, 0, 0],
        rotation=[0, 0, 0]
    ):
        """
        :param offset: pose parameters for the elementary primitive's
        ↪   initial position.
        :param rotation: pose parameters for the elementary primitive's
        ↪   initial rotation in Euler angles.
        """
        self.offset = offset
        self.rotation = rotation
        self.structure = None  # Mesh

    def translate(self, offset):
        """
        Translate the primitive according to the given values.
        """
        self.structure.translate(offset)

    def rotate(self, rotation):
        """
        Rotate the primitive (around the origin) according to the given
        ↪   values.
        """
```

```
25          self.structure.rotate(rotation)
26
```

**Example - Cylinder.** Below we show the codes for class cylinder as an elementary primitive. During initialization, it registers the parameters $R, h$ and creates a mesh of the cylinder.

```python
1  class Cylinder(Elementary_Primitive):
2      def __init__(
3          self, R, h,
4          offset=[0, 0, 0],
5          rotation=[0, 0, 0]
6      ):
7          """
8          :param R: radius of the cylinder
9          :param h: height of the cylinder
10         :param offset: offset (x, y, z) of the cylinder
11         :param rotation: rotation of the cylinder, represented via Euler
           ↪  angles (x, y, z)
           """
12         super().__init__(offset, rotation)
13         self.R = R
14         self.h = h
15         self.structure = create_mesh(
16             'cylinder',
17             radius=R, height=h,
18             offset=offset,
19             rotation=rotation
20         )
21
22
```

**Example - Cuboid.** We further provide the codes for class cuboid as another example of an elementary primitive, whose implementation is similar to that of the cylinder.

```python
1  class Cuboid(Elementary_Primitive):
2      def __init__(
3          self, sizes,
4          offset=[0, 0, 0],
5          rotation=[0, 0, 0]
6      ):
7          """
8          :param sizes: 3-dimensional sizes (x, y, z) of the cuboid
9          :param offset: offset (x, y, z) of the cuboid
10         :param rotation: rotation of the cuboid, represented via Euler
           ↪  angles (x, y, z)
11         """
12         super().__init__(offset, rotation)
13         self.sizes = sizes
14         self.structure = create_mesh(
15             'cuboid',
16             sizes=sizes,
17             offset=offset,
18             rotation=rotation
19         )
20
```

## H.2 ADVANCED PRIMITIVES

**Base Class.** For advanced primitives, we also implement the base class first. Besides the primitive's offset and rotation in 3D space, there are additional key functions. The functions cpa and dpa correspond to the first two procedural rules introduced in Sec. 3.4, *i.e.* CPA and DPA. The functions get_general_info and inherit_param_from together enable one advanced primitive to inherit features like overall dimensions from another during APA. And the function handle_exceptions is responsible for detecting and adjusting erroneous parameters of the

primitive and ensuring the structure's validity. Please refer to the following example for their implementations.

```python
class Advanced_Primitive:
    def __init__(
        self,
        offset=[0, 0, 0],
        rotation=[0, 0, 0]
    ):
        """
        :param offset: pose parameters for the advanced primitive's
        ↪  initial position.
        :param rotation: pose parameters for the advanced primitive's
        ↪  initial rotation in Euler angles.
        """
        self.offset = offset
        self.rotation = rotation
        self.structure_dict = {} # A registry for all the elementary
        ↪  primitives involved in the advanced primitive

    def make_structure(self):
        pass

    def translate(self, offset):
        """
        Translate the primitive according to the given values.
        """
        for structure in self.structure_dict.values():
            structure.translate(offset)

    def rotate(self, rotation):
        """
        Rotate the primitive (around the origin) according to the given
        ↪  values.
        """
        for structure in self.structure_dict.values():
            structure.rotate(rotation)

    def cpa(self):
        pass

    def dpa(self):
        pass

    def get_general_info(self):
        pass

    @classmethod
    def inherit_param_from(self, general_info_dict):
        pass

    def handle_exceptions(self):
        pass
```

**Example - GlobeBase_Star.** Below we give the implementation of a specific advanced primitive, *i.e.* the globe base in the style of a star, which is shown in Fig. 13. In the __init__ function, we declare attributes and functions and register the parameters.

```python
class GlobeBase_Star(Advanced_Primitive):
    default_parameters = {
        'stanchion_sizes': ...,
        'leg_sizes': ...,
        ...
```

```
6        }
7    def __init__(self,
8        stanchion_sizes, leg_sizes,
9        leg_tilt_angle, central_rotation,
10       number_of_legs,
11       offset=[0, 0, 0], rotation=[0, 0, 0]
12   ):
13
14       super().__init__(offset, rotation)
15       self.stanchion_sizes = stanchion_sizes
16       self.leg_sizes = leg_sizes
17       self.leg_tilt_angle = leg_tilt_angle
18       self.central_rotation = central_rotation
19       self.number_of_legs = number_of_legs
20       self.offset = offset
21       self.rotation = rotation
22       self.handle_exceptions()
23       self.make_structure()
24
```

In function `make_structure` we give the detailed steps of constructing the structure. Note that the connectivity relationships between the elementary primitives are already implicitly embedded in the process. Fig. 13 illustrates the structure and the effects of different parameters.

```
1    # Continue Above
2    def make_structure(self):
3        stanchion_offset = [
4            0,
5            -self.stanchion_sizes[1] / 2,
6            0
7        ]
8        stanchion_rotation = [
9            0,
10           self.central_rotation,
11           0,
12       ]
13       self.structure_dict['stanchion'] = Cylinder(self.stanchion_sizes,
         ↪   stanchion_offset, stanchion_rotation)
14       for leg_idx in range(self.number_of_legs):
15           central_rot = self.leg_sizes[2] / 2 *
             ↪   cos(self.leg_tilt_angle) * sin(2 * pi /
             ↪   self.number_of_legs * leg_idx)
16           tilt_adduction_x = self.leg_sizes[2] / 2 *
             ↪   cos(self.leg_tilt_angle) * sin(central_rot)
17           tilt_adduction_z = self.leg_sizes[2] / 2 *
             ↪   cos(self.leg_tilt_angle) * cos(central_rot)
18           offset_y = -self.stanchion_sizes[1] + self.leg_sizes[1] / 2 -
             ↪   self.leg_sizes[2] * sin(self.leg_tilt_angle) / 2
19           offset_z = tilt_adduction_z * cos(self.central_rotation) +
             ↪   tilt_adduction_x * sin(self.central_rotation)
20           leg_i_offset = [
21               tilt_adduction_z * sin(self.central_rotation) -
                 ↪   tilt_adduction_x * cos(self.central_rotation)
22               offset_y,
23               offset_z,
24           ]
25           leg_i_rotation = [
26               self.leg_tilt_angle,
27               -central_rot,
28               0
29           ]
30           self.structure_dict['leg_%d' % leg_idx] =
             ↪   Cuboid(self.leg_sizes, leg_i_offset, leg_i_rotation)
31
32       self.rotate(self.rotation)
```

```
            self.translate(self.offset)
```

The function `cpa` applies perturbations to all the continuous parameters of the primitive, whereas `dpa` changes the discrete parameters (*e.g.* the number of legs in this case). Both functions automatically check for and correct the exceptions with the help of `handle_exceptions`, and then update the structure with `make_structure`. Fig. 13 also indicates examples of such alterations. The function `handle_exceptions` operates by actively checking for parameter combinations that could lead to collisions and adjusting erroneous parameters.

```
# Continue Above
def cpa(self):
    apply_perturbation(self.stanchion_sizes)
    apply_perturbation(self.leg_sizes)
    ...

    self.handle_exceptions()
    self.make_structure()

def dpa(self):
    self.number_of_slats = random_choice(
        range(self.maximum_num_legs)
    )
    self.handle_exceptions()
    self.make_structure()

def handle_exceptions(self):
    while self.leg_sizes[2] * sin(self.leg_tilt_angle) <
     ↪   self.stanchion_sizes[0]:
        increase_value(self.leg_sizes[2])
        reduces_value(self.leg_tilt_angle)
    # gradually increase the sizes of the legs and reduce the tilt
     ↪   angle until they together broaden outer edge of legs to form
     ↪   a stable frame

    while 2 * self.stanchion_sizes[0] > self.leg_sizes[0]:
        increase_value(self.leg_sizes[0])
        reduces_value(self.stanchion_sizes[0])
    # gradually increase the sizes of legs and reduce the radius of
     ↪   stanchion until legs are not blocked by stanchion.

    ...
```

For APA, we introduce functions `get_general_info` and `inherit_param_from`. The original primitive uses the former one to record its general information in a dictionary, which contains its basic dimensions at a macro level. Then, the replacement primitive can receive the dictionary with the latter one to determine its dimensions accordingly.

```
# Continue Above
def get_general_info(self):
    """
    :return: A dictionary listing the general information of the
     ↪   primitive indexed by keywords. These keywords are shared
     ↪   among advanced primitives that represent a component at the
     ↪   same hierarchy
    """
    general_info_dict = {
        'outer_dimension_y' = self.stanchion_sizes[1] +
         ↪   self.leg_sizes[0] * cos(self.leg_tilt_angle)
        'outer_radius' = self.leg_sizes[0] * sin(self.leg_tilt_angle)
        'stanchion_radius' = self.stanchion_sizes[0]
        'stanchion_height' = self.stanchion_sizes[1]
        'leg_length' = self.legs_sizes[0]
```

```
12            ...
13        }
14        return general_info_dict
15
16    @classmethod
17    def inherit_param_from(cls, general_info_dict):
18        """
19        Inherit key parameters from the general_info_dict of another
           ↪    advanced primitve
20        """
21
22        # begin with default parameters
23        inherited_parameters = copy.deepcopy(
24            cls.default_parameters
25        )
26
27        # the height of the stiles are inherited if the other advanced
           ↪    primitive also features 'inner_dimension_y'
28        if 'stanchion_radius' in general_info_dict:
29            inherited_parameters['stanchion_sizes'][0] =
               ↪    general_info_dict['stanchion_radius']
30
31        # some parameters are calculated instead of directly inherited
32        if 'outer_radius' in general_info_dict \
33                and 'leg_length' in general_info_dict:
34            inherited_parameters['leg_tilt_angle'] =
               ↪    acos(general_info_dict['outer_radius'] –
               ↪    general_info_dict['leg_length'])
35        ...
36        return inherited_parameters
37
```

More advanced primitives for different types of the globe ball, bracket and base can be defined in a similar way with essential parameters, constructors, functions such as `cpa`, `dpa`, *etc.*

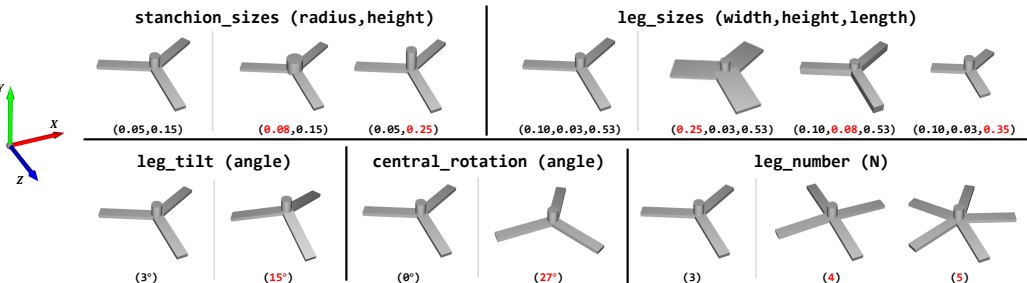

Figure 13: Illustrations of the structure and the effects of different parameters for `GlobeBase_Star` corresponding to its structure program. Each cell consists both the original structure and structures with an altered parameter marked in red. These illustrations also indicate examples of CPA and DPA.

## H.3 OBJECTS

**Example - Globe.**  Now, we show the codes for globe as an example of representing objects with structure programs. The `__init__` function receives multiple configurations and then uses them to initialize the components of the object. Each configuration is a dictionary that specifies a primitive template and its parameters for a hierarchical component. As for structure manipulations, CPA and DPA are implemented by directly invoking the corresponding functions of the object's components. For APA, we change the primitive of certain components and obtain its parameters with the help of `get_general_info` and `inherit_param_from` as aforementioned. And similar to advanced primitives, the function `handle_exceptions` is used to ensure the validity of the structure.

```python
class Globe:
    def __init__(self, ball_cfg, bracket_cfg, base_cfg):
        """
        :param ball_cfg: ...
        :param bracket_cfg: ...
        :param base_cfg: {
            'cls': Advanced_Primitive,
            'param': Dict
        }
        """

        self.ball_structure = ...
        self.bracket_structure ...
        self.base_structure = eval(base_cfg['cls'])(**base_cfg['param'])

    def move_to_pose(rotation, offset):
        self.ball_structure.rotate(rotation)
        self.ball_structure.translate(offset)
        ...

    def cpa(self):
        self.ball_structure.cpa()
        self.bracket_structure.cpa()
        self.base_structure.cpa()

        self.handle_exceptions()

    def dpa(self):
        self.ball_structure.dpa()
        self.bracket_structure.dpa()
        self.base_structure.dpa()

        self.handle_exceptions()

    def apa(self):
        ...

        new_base_type = get_random_component_name('globe', 'base')  #
        ↪  Randomly select a new base type from the advanced primitives.
        base_general_info = self.base_structure.get_general_info()
        new_base_type_parameters = eval(new_base_type).inherit_from(
            base_general_info
        )
        self.base_structure = eval(new_base_type)(
            **new_base_type_parameters
        )

        ...

        self.maintain_connectivity()
        self.handle_exceptions()

    ...
```

## I  ADVANTAGES BEHIND THE DESIGN, LIMITATIONS AND FUTURE WORK

### I.1  SCALABILITY OF DESIGNING PRIMITIVE TEMPLATES

Primitive templates are fundamental for Arti-PG, and we have already provided more than 200 templates in the toolbox to cover 26 categories of commonly seen articulated objects. We also find that there may be users who want to customize their own templates to satisfy their needs, and here we show how the elaborate design of primitive templates can mitigate the costs to create new ones.

As stated in Sec. 3.2, we propose a two-tier design of primitive templates. Elementary primitive templates, representing the basic and general geometric shapes, are first defined from scratch. Then advanced primitive templates can be defined upon elementary ones instead of from scratch, to represent the diverse structures of articulated objects. Therefore, 1) with pre-defined elementary ones, scaling up the advanced ones is practically convenient at the program level, and 2) many advanced templates are reusable across object categories (e.g. a template of handle can be used in window, door, fridge, etc.), indicating that scaling up the number of object categories covered by Arti-PG is also convenient. To take a step further, as the scale of advanced primitives goes larger, the scaling of object categories can be easier.

We will make the primitive templates that we have already created publicly available in the Arti-PG toolbox for researchers to use directly. If someone needs to define primitive templates for a new category, he/she can leverage the ones we provided, avoiding the burden of designing from scratch. We will also continue to extend our work to include more object categories and share the newly defined primitive templates with the community, making our work stronger.

## I.2 Advantages over Collecting and Annotating more Real Objects

To address the data scarcity issue of articulated objects, *i.e.* lack of both object data and annotations for various articulated object understanding tasks, there are currently two possible ways: (1) collecting and annotating more real objects (abbreviated as CARO), and (2) procedurally generating objects (our approach). For CARO, the obstacles are i) collecting real articulated objects and ii) providing different types of annotations for each object.

Regarding obstacle i), due to the complex structure of articulated objects, the object collection process is difficult and time-consuming. For reference, the average time to collect a CAD articulated object is more than 120 minutes and the cost is more than \$100 (Liu et al., 2022). The average time to scan an articulated object is 20 minutes and an additional 15 minutes are needed to fix imperfect meshes from the scan (Liu et al., 2022; Geng et al., 2023). As scanning requires purchasing objects, the cost can be high, especially for categories like electrical appliances and furniture (Liu et al., 2022). Further, both collection practices require experts, *i.e.* who are capable of designing CAD models, labeling the URDF or using a scanner (Liu et al., 2022).

As for obstacle ii), given the large number of articulated object understanding tasks as stated in Sec. 2.2, many different types of annotations need to be annotated on these objects to enable training for these tasks. For reference, the average time to annotate part semantics for a 3D object is about 8 minutes (Mo et al., 2019), and to annotate part pose is about 10 minutes (Geng et al., 2023).

In summary, at least about an hour and tens of dollars are cost on average for only one object in CARO. Therefore, CARO is expensive and time-consuming.

In our approach, the design of primitive templates and structure program annotation requires human effort. The average time to design primitive templates to cover an object category is about 6 hours, which is a once-and-for-all effort. Additionally, the structure program annotation step takes about 6 minutes per object. Further, as we will make these codes and data publicly available as a toolbox, such efforts are free for users in the community. This substantially demonstrates the efficiency and scalability of Arti-PG, as well as its superiority compared to CARO.

## I.3 Limitations

In this paper we propose a novel and effective procedural approach for synthesizing articulated objects for network training. However, despite the great variations in the structure of the synthesized objects, there is still room for diversifying the geometric details. In addition, Arti-PG currently focuses on 3D visual features and is not coupled with rgb features like color and texture. We will take these points as our future work to better alleviate the data scarcity issue.

## I.4 Future Work

Our current approach is an exploration in the context of scarcity of 3D articulated objects. We believe that in the future, when 3D articulated objects are no longer scarce, abundant data will unleash greater potential for using Arti-PG toolbox to generate object spatial structures. A possible way is

to first use a generative model to learn the distribution of parameters from the structure program annotation of abundant real articulated objects, and then use the distribution to infer parameters of the primitives to generate new instances. We will consider this as our future work. We will also continue working on extending Arti-PG toolbox to more object categories and tasks.

