# OpenReview forum: "Arti-PG: A Procedural Toolbox to Synthesize Large-Scale and Diverse Articulated Objects with Rich Annotations"
_ICLR.cc/2025/Conference — ICLR 2025 Conference Withdrawn Submission_

### Official Review · Reviewer_8Yrx · 2024-10-26

**Soundness:** 2
**Presentation:** 3
**Contribution:** 2
**Rating:** 5
**Confidence:** 3

**Summary:**

The manuscript proposes an articulated object synthesis framework, Arti-PG. The method is designed based on the insight that the articulated object is structural and can be viewed as created via basic primitives in a structural way. Arti-PG proposes to decompose the generation process into global structure creation by manipulating structures and generating detailed local geometry by fitting and aligning the point clouds. With this hierarchical generation philosophy, Arti-PG proposes a two-stage generation pipeline, together with a labeling and annotation process. Arti-PG can successfully create large amounts of articulated objects with diverse shapes. Experiments demonstrate that models trained using these generated articulated objects can perform better than those trained with a small amount of data. Concerns lie in the quality of the generated shapes and the efficiency of the system.

**Strengths:**

- Hierarchical shape modeling and generation is a smart and effective representation. The paper is well-motivated. Leveraging the structural prior and cross-instance local geometry relations, Arti-PPG proposes a promising way for generating articulated objects.

- The development of an articulated object generation toolbox for various applications. Generating articulated objects in a convenient and efficient manner is quite crucial for many downstream tasks that have a high demand for large-scale articulated object data.

- Extensive experiments with real-world evaluations. The authors present a wide range of generated results to demonstrate the effectiveness of the method and the quality and diversity of the generated objects. Besides and various downstream applications are included in the experiments, demonstrating the value of a large scale of diverse articulated objects.

- Detailed visualizations including videos are provided in the Appendix and Supp. Besides, very detailed implementations covering pseudo code are provided in the supplementary material.

**Weaknesses:**

- Reasonability of the method.  The quality of the generated results is not naturally guaranteed in the method. The design in generating the local details cannot make sure the validity and the quality of the generated object. For instance, simply leveraging this method without a human-in-the-loop design can easily result in invalid outputs, e.g., parts collide with each other in the articulated motion.

- The efficiency and the applicability. The method is semi-autonomous and relies on human efforts. Therefore it is indeed questionable whether the method can help with scaling up the articulated object dataset in a reasonable time budget.

- Quality and diversity. Although the authors have shown appealing generation results, it seems that many instances are still limited to the original articulated objects available in their considered datasets. Thus it is questionable w.r.t. the sample diversity. Creating objects with limited diversity using a method that makes it hard to generate brand-new objects would downweight the value of the method.

**Questions:**

Can the generation method itself ensure the physical fidelity of the generated objects?

---

### Official Review · Reviewer_ivN2 · 2024-11-01

**Soundness:** 1
**Presentation:** 1
**Contribution:** 2
**Rating:** 3
**Confidence:** 4

**Summary:**

-	Observing the data sparsity issue for collecting articulated objects, this paper proposes to use procedural generation to assist the creation of 3D articulated objects with various annotations.
-	By taking an existing object as input, the system represents an articulated object as a combination of a macro spatial structure and a micro geometric detail. A structure program is used to specify the primitive geometry and connectivity for each part as a descriptor of the general structure of the input object. Then the geometric details is describe as the deformation from the primitives to the original surface by finding the point-correspondence.
-	Once representing the object in the program, it takes two steps to synthesize a new 3D articulated object: use mathematical rules to randomize the structure first, and then recover the geometry with point-wise correspondence. Following the 3D synthesis, a series of mathematical rules is applied to annotate the object automatically. All these components are integrated into the proposed Articulated Object Procedural Generation toolbox (ArtiPG-toolbox).

**Strengths:**

-	This paper identifies an important data gap in the field of articulated object modeling and contributes to densifying and diversifying the synthetic data for articulated objects.
-	This paper designs a practical system to automate the process of creating alternative versions of the input object with various annotations.
-	This paper shows a user-friendly interface to help users to synthesize new objects using the system.

**Weaknesses:**

-	**Paper writing should be improved**. The paper writing is overall difficult to follow, mainly due to 1) the repeating contents in the introduction and related work sections that can be better structured; 2) the explanation in the method and experiment sections is not specific enough to understand the details; 3) some content is inconsistent throughout the paper, e.g., the number of the objects collected using Arti-PG (3096 in introduction, 2133 in line 394).
-	**Unclear contribution**. It is unclear what the main contributions are claimed and demonstrated in this work. Based on my understanding, the proposed system is helpful to automate the augmentation process of the existing datasets, but it is not really creating new objects at scale as it is essentially using the existing datasets as a library to composite more objects in a combinatorial way under certain assumptions and with lots of manually crafted rules.
-	**Experiment purpose is vague**. It is unclear what the experiment section is trying to validate. Why the three tasks are chosen to do the benchmark? Why only the PointWOLF is reported as the baseline? For the data itself, how is the realism of the synthesized objects evaluated and guaranteed?
-	**Insufficient discussion on the assumption and limitation**. There are many assumptions made in constructing the Arti-PG toolbox but never discussed. It is also unclear what this system is good at and what its limitation is.

**Questions:**

**Questions**
- About data annotation: what annotations are provided for the data? Where are the annotations from? Is the affordance manually crafted for each primitive?
- About the exception handling module: how does it work exactly? How can it make sure the parts are geometrically and kinematically plausible?
- About Discrete Parameter Alteration (DPA): Does the change to the part only affect the part itself? Would other related parts adapt accordingly? For example, if I want to change a cabinet originally with one door to two doors, would the cabinet body/frame adjust to make more space for the additional door?

**Suggestions**
- As mentioned in the `Weaknesses` section, this work can benefit from explicitly summarizing the main contributions, which also helps readers to understand other sections better.
- This work can be better contextualized by re-organizing the related work section with more references and discussion on the connections with the prior work, e.g. incorporating the subsection of "articulated object synthesis", and combining sections 2.1 and 2.3 into one block.

**Some closely related references that are missing**:
- Lei, Jiahui, Congyue Deng, William B. Shen, Leonidas J. Guibas, and Kostas Daniilidis. "Nap: Neural 3d articulated object prior." Advances in Neural Information Processing Systems 36 (2023): 31878-31894.
- Liu, Jiayi, Hou In Ivan Tam, Ali Mahdavi-Amiri, and Manolis Savva. "CAGE: Controllable Articulation GEneration." In Proceedings of the IEEE/CVF Conference on Computer Vision and Pattern Recognition, pp. 17880-17889. 2024.
- Luo, Rundong, Haoran Geng, Congyue Deng, Puhao Li, Zan Wang, Baoxiong Jia, Leonidas Guibas, and Siyuang Huang. "PhysPart: Physically Plausible Part Completion for Interactable Objects." arXiv preprint arXiv:2408.13724 (2024).
- Liu, Jiayi, Manolis Savva, and Ali Mahdavi-Amiri. "Survey on Modeling of Human-made Articulated Objects." arXiv preprint arXiv:2403.14937 (2024).

---

### Official Review · Reviewer_An9k · 2024-11-04

**Soundness:** 3
**Presentation:** 2
**Contribution:** 2
**Rating:** 5
**Confidence:** 3

**Summary:**

The paper proposes a toolbox for annotating 3D articulated objects. It consists of 3 components: (i) descriptions of articulated objects by a generalized structure program and point correspondences; (ii) variations on the structure program to synthesize diverse new articulated objects; (iii) additional annotations such as affordance and semantics.

**Strengths:**

- Datasets for articulated objects with high-quality annotations are indeed inadequate nowadays. The paper is addressing this important problem with an effective framework for data annotation.

**Weaknesses:**

- To my understanding, this paper is positioned as a dataset and benchmark paper, with its main contribution being a toolbox for data annotation. If this understanding is correct, I would expect more data analysis:
	- Simply 3096 objects does not sound enough for a dataset contribution, but it may also be reasonable given the complex data annotation for 3D articulated objects. How does this number compare to existing datasets?
	- What are the qualities, distributions, and other features of the dataset (or in general, the annotated data with the toolbox)? It would be good to show more statistics about that.
- The writing of the paper makes me feel a bit hard to follow. For example, it would be good to show an overview figure of what the procedural generation process is like. Also, why is the sequence of operations "procedural"? It would be really good to have more high-level explanations, including figures, in addition to the technical details, to help readers better understand the overall framework.

**Questions:**

- As discussed in weaknesses, I may want to see more analysis of the dataset, such as its data statistics.

---

### Official Review · Reviewer_5aE3 · 2024-11-05

**Soundness:** 2
**Presentation:** 2
**Contribution:** 2
**Rating:** 6
**Confidence:** 4

**Summary:**

This paper introduces Articulated Object Procedural Generation toolbox (Arti-PG toolbox) that speeds up generating 3D articulated objects with part annotations. This toolbox contains (1) structure programs with correspondence to surface point cloud, (2) procedural manipulation (3) mathematical knowledge description. Experiments demonstrate that the data generated by Arti-PG can improve models' performance on both vision and robotic manipulation tasks.

**Strengths:**

1. This paper is well-motivated -- many tasks related to articulated objects face the problem of insufficient amount of data. To address this problem, this paper proposes novel components and procedures that generate 3D articulated shapes with large variations and detailed annotations.

2. The paper is well-written and easy to follow. Implementation details have been sufficiently provided in both main paper and supplementary. So this paper should be reproducible.

3. Experiments for both vision and robotic tasks (part segmentation, part pose estimation, point cloud completion and object manipulation) have shown that training on data generated by Arti-PG toolbox can indeed improve the performance, which proves the usefulness of this toolbox and its data.

**Weaknesses:**

1. The paper mostly shows articulated objects with one/two joint with very few exceptions (boxes with four joints). Is there any limitation on generating objects with much more joints (e.g., 10)?

2. Although advanced primitive is available, the generated shapes still seem to lack fine geometric details, which typically exist in real-world objects. Without these details, I was wondering if the authors have any thoughts on how this would affect the sim-to-real gap.

3. How does this method handle some big structural variations within a category, e.g., chair with four straight legs vs. swivel chair.

**Questions:**

If the parts have some complicated geometries (e.g., concave shapes), then the joint does not necessarily ly on the boundary of the OBB. In this case, the parts at the two ends of the joint may not be connecting well or may have some unwanted intersection. What are the authors' thoughts on this?

---

### Note · Authors · 2024-11-15

I have read and agree with the venue's withdrawal policy on behalf of myself and my co-authors.